# Compression is all you need for Controllably Efficient Language Models

## Abstract

The substantial inference costs of attention in transformers motivated the development of efficient sequence mixers: namely sparse and sliding window attention, convolutions and linear attention. Although these approaches result in impressive reductions in inference costs, they often trade-off with quality, specifically in-context recall. Moreover, apriori fixing this quality-cost tradeoff in an architecture means being suboptimal from the get-go: some downstream applications might fundamentally require more memory for in-context recall, while other tasks may require lower latency and memory.

To address above issues, we propose a conceptually simple *meta*-sequence mixer: **C**ompress & **A**ttend **T**ransformer (CAT). CAT decodes chunks of tokens by *attending* to compressed chunks of the sequence so far. Both compression and decoding from compressed chunks can make use of any existing sequence mixers. Notably, compression results in decoding from a reduced sequence length that yields compute and memory savings, while choosing a particular chunk size trades-off quality for efficiency. Importantly, training CAT with multiple chunk sizes at once, unlocks control of quality-efficiency trade-offs directly at test-time without any retraining, all in a single adaptive architecture.

We instantiate CAT with most basic choice of dense attention as mixer and demonstrate it suffices to match or surpass multiple popular efficient baselines on long-context recall benchmarks at similar inference costs, all using a **single** model only. Further, CAT performs competitively in long-context understanding benchmarks, while being $1.4 - 3\times$ faster and requiring $2 - 9\times$ lower total memory usage than a dense transformer.

[1]Anonymous Institution, Anonymous City, Anonymous Region, Anonymous Country. Correspondence to: Anonymous Author <anon.email@domain.com>.

Preliminary work. Under review by the International Conference on Machine Learning (ICML). Do not distribute.

## 1. Introduction

Transformers (Vaswani et al., 2017) rely on self-attention (Bahdanau et al., 2014) to power large language models (LLMs). However, the cost of decoding with self-attention grows quadratically in compute and linearly in memory with sequence length. This makes transformers expensive to deploy in a world where inference cost dominates in the long run due to reasoning and inference-time scaling[1](Timbers, 2023; Ord, 2025). Inference costs are often driven by memory due to hardware are memory-bound (Gholami et al., 2024).

These costs motivated efficient alternatives in the community: while approaches like sparse and sliding window attention (Child et al., 2019; Zaheer et al., 2020; Jiang et al., 2023) heuristically restrict the tokens being attended to, approaches like linear attention (Katharopoulos et al., 2020; Arora et al., 2024a; Dao & Gu, 2024; Yang et al., 2025b) rely on a fixed-size recurrent state with complex state update rules, to enable constant compute and memory costs. However, restricting attention to tokens apriori or using fixed-size recurrent states, that have problems managing information over long sequences, hurt in-context recall performance (Arora et al., 2024a; Jelassi et al., 2024; Wen et al., 2024). Making these approaches performant requires careful composition with dense attention at specific layers to create hybrids, making the design process cumbersome especially at scale (Waleffe et al., 2024; Wang et al., 2025). Enabling efficiency by recursively compressing the sequence can avoid fixed-memory bottlenecks and heuristic restrictions (Rae et al., 2020; Chevalier et al., 2023), but sequential computations in these approaches makes the training slow and optimization difficult (Geiping et al., 2025).

Moreover, existing approaches do not account for the differences in compute and memory requirements across diverse downstream tasks. For example, writing short email replies does not require strong in-context recall performance and usage of linear attention can be sufficient; but code auto-completion demands accurate recall of function names from the entire code repository in the context, where more memory and compute of dense attention may be preferred. The existing approaches for efficiency *fix* the flops/memory us-

[1]in fact, the cost of training can break even with inference cost in *just* a few weeks (Timbers, 2023)

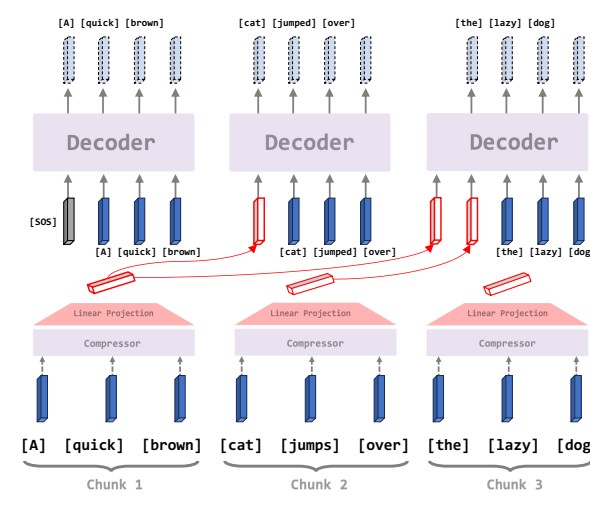

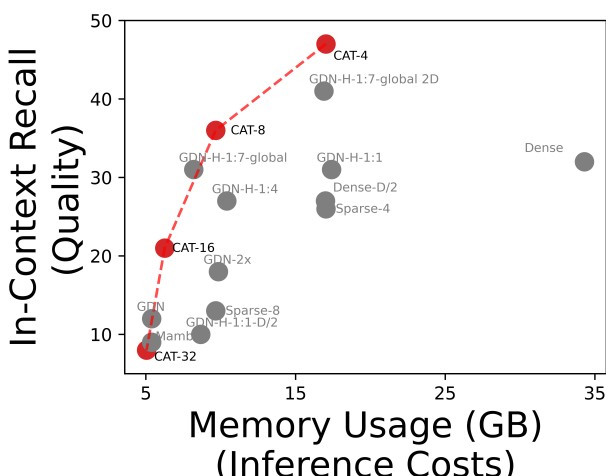

*(a)* **The Compress and Attend Transformer (CAT) architecture.** CAT chunks up a sequence of length $N$ into $N/C$ chunks of $C$ tokens (illustrated for $C = 3$). Each chunk is parallelly compressed into a chunk representation. CAT then decodes each chunk by attending to past chunk representations. Observe the reduced sequence length in the decoder. Chunk size in CAT acts as a ***knob***, offering test-time control of quality-efficiency trade-offs, where higher chunk sizes result in increased efficiency.

*(b)* CAT **unlocks test-time control of quality-inference cost trade-offs**: a **single** adaptive CAT model (**red** dots) matches or surpasses efficient approaches at comparable inference costs on real-world in-context recall tasks. We compare against popular baselines: **across 12 models**, all having diverse model configurations, parameter counts (∼300M to ∼820M) and **varying inference costs** for fair and broad evaluation. When latency-matched, a single CAT is competitive with most models (Appendix Fig. 5).

*Figure 1.* Overview of CAT.

age before training. This means if at test time, a problem demands a higher budget for better performance, a whole new model needs to be trained. Training multiple models with different tradeoffs is one way to tackle this problem but repeating this for every downstream task can become quickly prohibitive.

To address above issues, this paper asks two research questions: (i) can we develop an approach where the per-token inference cost is controllable at test time? and (ii) can we instantiate such an approach using simple, standard components, without careful hybridization or custom kernels, that performs comparably or better than specialized approaches at similar inference costs?

For the first question, we propose a *meta*-sequence mixer – a conceptually simple arrangement where one sequence mixer compresses the sequence, and another attends to this compressed sequence while decoding. We term this approach: **C**ompress & **A**ttend **T**ransformer (CAT). Concretely, CAT compresses chunks of tokens in parallel into a shorter sequence using a compressor, which a decoder then attends to while autoregressively modeling the tokens in the latest chunk. The compression and decoding is parallel over tokens during training, meaning there is no recurrence along the sequence, enabling **end-to-end scalable training**. When the cost of the sequence mixer depends on the length of the sequence, decoding from the reduced sequence length due

to compression enables **compute and total memory savings** during inference. This reduction in-turn allows the use of more parameters in CAT, thereby improving model quality while still maintaining same or lower inference cost.[2] Importantly, training CAT across multiple chunk sizes at once **unlocks control of quality-inference costs trade-offs directly at test-time** without any retraining.

For the second question, we make the most vanilla choice and instantiate CAT with both the compressor and decoder as simple dense transformer black-boxes (see Figure 1a). Since dense attention is a staple and well-developed, training and inference can be done efficiently with existing machinery. We find CAT with simple dense attention suffices to match or outperform several popular alternatives without custom kernels or careful choices across varying inference budgets. Notably, this is achieved with a **single** CAT model whose inference cost can be controlled on the fly (see Figure 1b).

Finally, we can use the same principles and instantiate CAT as a drop-in layer in any architecture, where it can be mixed and matched with other sequence mixers (e.g. linear attention) to bring test-time control wherever needed.

To summarize:

---

[2]similar in vein to deployed flagship models (Yang et al., 2025a; Agarwal et al., 2025) that are all parameterized as Mixture-of-Experts (MoEs) (Shazeer et al., 2017), where additional parameters brings improvements in quality without changing inference costs.

- We introduce the controllably efficient meta-sequence mixer: CAT. Adjusting a single knob (chunk size) at test-time controls quality-efficiency trade-offs, allowing a single CAT model to *interpolate* between the performance of dense transformer and efficient alternatives without any retraining.

- We provide a parallel and scalable implementation for training CATs (we scale from 90M to 1B parameters) and an efficient pure PyTorch implementation for generation that does not require any custom CUDA or Triton kernels, unlike most efficient approaches.

- We demonstrate that a **single** adaptive CAT model:
  - outperforms many popular efficient baselines including hybrid architectures on real-world long-context recall tasks across different inference budgets.
  - performs competitively in long-context modeling and understanding benchmarks
  - matches the dense transformer on language modeling while having lower inference costs (being $1.4 - 3\times$ faster and using a $2 - 9\times$ smaller total memory footprint).
  - surpasses, interestingly, even the dense transformer on real-world recall tasks using the most performant setting (CAT-4) while still consuming lower inference costs ($1.5\times$ faster and $2\times$ memory efficient), akin to MoEs (Shazeer et al., 2017).

- We provide preliminary results highlighting extensibility of CAT: the decoder can be make use of other sequence mixers (e.g., hybrids of attention recurrences), and CAT itself can serve as a drop-in replacement layer in any architecture.

## 2. Compress and Attend Transformers (CATs)

**Compression and decoding.** CAT is a meta–sequence mixer that uses one sequence mixer to compress chunks of a sequence and another to decode within each chunk given compressed representations. For simplicity, we instantiate sequence mixers for compression and decoding as vanilla dense attention, and showcase how **far** simple choices with CAT can go, yielding a competitive and test-time controllable architecture.

Concretely, given a sequence $\mathbf{x} = (x_1, x_2, \ldots, x_N)$ of $N$ tokens, we split the sequence into chunks $(\mathbf{c}_1, \mathbf{c}_2, \ldots, \mathbf{c}_{N_C})$ containing $C$ tokens each, such that $\mathbf{c}_i = (x_{C \cdot i+1}, \ldots, x_{C \cdot i+C}) = (\mathbf{x}_{i,1}, \ldots \mathbf{x}_{i,C}) = \mathbf{x}_{i,:}$, where $\mathbf{x}_{i,:}$ indexes the $i$-th chunk of $C$ consecutive tokens (numpy array slicing). CAT compresses each chunk $\mathbf{c}_i$ using the compressor $f_\theta$ into chunk representations. The compressor $f_\theta$ is a dense bidirectional transformer with hidden size

$D_f$, followed by a linear projection to $D_g$. This leads to a compressed chunk representation $f_\theta(\mathbf{c}_i) \in \mathcal{R}^{D_g}$. That is:

$$\{x_1, \cdots x_N\} \xrightarrow{\text{chunk}} \{\mathbf{c}_i\}_{i=1}^{N_c} \xrightarrow{\text{compress}} \{f_\theta(\mathbf{c}_i)\}_{i=1}^{N_c}$$

After compression, CAT decodes the original sequence $\mathbf{x}$ from the compressed chunk representations $\{f_\theta(\mathbf{c}_i)\}_{i=1}^{N_C}$ using a decoder $g_\theta$, which is a causal dense transformer, having hidden size $D_g$, matching the linear projection from the compressor. CAT decodes chunks autoregressively, where to decode each token $\mathbf{x}_{i,j}$ in a chunk $\mathbf{c}_i$, the decoder takes as input the previous tokens $\{\mathbf{x}_{i,<j}\}$ in chunk $\mathbf{c}_i$ and the past chunk representations $\{f_\theta(\mathbf{c}_1), \ldots, f_\theta(\mathbf{c}_{i-1})\}$. Formally, the predictive distribution $p_\theta$ for the tokens in chunk $\mathbf{c}_i$ is defined as:

$$p_\theta(\mathbf{c}_i \mid \mathbf{c}_{i-1} \cdots \mathbf{c}_1) =$$
$$\prod_{j=1}^{C} g_\theta \left( \underbrace{\mathbf{x}_{i,j}}_{j^{\text{th}} \text{ token in chunk } \mathbf{c}_i} \middle| \begin{array}{c} \underbrace{\mathbf{x}_{i,j-1}, \ldots, \mathbf{x}_{i,1},}_{\text{previous tokens in chunk } \mathbf{c}_i} \\ \underbrace{f_\theta(\mathbf{c}_{i-1}) \cdots f_\theta(\mathbf{c}_1)}_{\text{past chunk representations}} \end{array} \right) \quad (1)$$

When decoding cost depends on sequence length, CAT reduces the amount of compute and memory required by using compressed chunk representations; the larger the chunk size the larger the reduction in memory and compute.

During training, the compression and the decoding happens in parallel for all tokens in the sequence because compression of a chunk does not depend on earlier chunks. This choice allows the entire CAT model to be efficiently trained end-to-end with the standard next-token prediction loss. The end-to-end training ensures that CATs *learn what to retain* in their compressed chunk representations rather than relying on fixed attention patterns or complex state update rules.

**Training for test-time control of inference costs.** Changing the chunk size in CATs trades-off quality for compute and memory efficiency. Training CAT with multiple chunk sizes renders a single adaptive model whose compute-memory budget can be adjusted directly at test-time without any retraining. We uniformly sample a chunk size $C$ at each training iteration, and pass in a *learnable* indicator token to CAT to indicate which chunk size it is currently operating at. The compressed tokens are separated from the uncompressed ones in the decoder using a marker token shared across different chunk sizes. After training, one can use the same CAT model at different compute/memory budget by just changing the indicator token at test-time. App. C.4 provides further detail.

**CAT as a layer.** using the same principles discussed above, CAT can be instantiated as a modular layer that can be mixed-and-matched with existing approaches and inserted in any architecture, unlocking controllable costs there and new hybrid designs. In a layer form: simple linear projection

can be the compressor, and a sequence mixer itself (say dense attention) can be the decoder. App. B.10 provides preliminary results, and we leave full exploration to future work.

### 2.1. How to implement fast and scalable CATs

When both compressor and decoder are dense transformers, CAT admits a pure PyTorch implementation for scalable training and fast generation requiring no custom CUDA or Triton kernels. We describe this approach below.

**Fast and Parallel Compression.** Compression of chunks of tokens is efficient and can be executed in parallel, for instance by using `torch.vmap`, to produce $\{f_\theta(\mathbf{c}_i)\}$ for all chunks $\mathbf{c}_i$. This costs a total of $O(\frac{N}{C} \cdot C^2) = O(NC)$ in self-attention compute, rather than $O(N^2)$.

**Naive and Slow Training.** For training the decoder, a naive implementation can lead to slower training. To compute logits for tokens in chunk $\mathbf{c}_i$, that is computing $g_\theta(\mathbf{c}_i \mid f_\theta(\mathbf{c}_1) \cdots f_\theta(\mathbf{c}_{i-1}))$ in parallel can be non-trivial. Since, for chunk $\mathbf{c}_i$, the number of past chunks varies, making shapes variable and as a result, harder to parallelize the computation of logits. One could employ a python loop and compute logits for every chunk sequentially, but that would be slow and would not scale. Padding to make shapes constant to allow parallelism would make things worse by increasing wasteful computations. In fact, even if one bypasses varying shapes problem and manages to compute logits for every chunk in parallel, the total self-attention operations in the decoder would scale as $O(\sum_{i=1}^{N_c} (i + C)^2) = O((\frac{N}{C})^3)$, that is cubic in sequence length. Thus, even the ideal parallel approach for training will not scale, despite the simplicity of CAT.

**Parallel and Scalable Training.** To make training scalable in CATs, we observe that in computing logits for every chunk $\mathbf{c}_i$, one calculates exactly the same key-value vectors for the representation $f_\theta(\mathbf{c}_j)$ in the decoder transformer, where $j < i$. This means that computation is duplicated. We exploit this observation in training CATs.

On a high-level, we implement this observation by modifying the original chunked sequence $\mathbf{x} = \{\mathbf{c}_1, \ldots \mathbf{c}_i \ldots\}$ to $\{\mathbf{c}_1, f_\theta(\mathbf{c}_1), \mathbf{c}_2, f_\theta(\mathbf{c}_2), \ldots \mathbf{c}_i, f_\theta(\mathbf{c}_i) \ldots\}$, that is we insert compressed representations of the chunk after the chunk of tokens itself. Now, we pass this sequence into the decoder during training, with a custom attention mask (App. Figure 9) that allows a token in chunk $\mathbf{c}_i$ to attend to previous tokens within that chunk and *only* to previous chunk representations, which would be $f_\theta(\mathbf{c}_{i-1}), f_\theta(\mathbf{c}_{i-2}) \ldots f_\theta(\mathbf{c}_1)$. Any token in chunk $\mathbf{c}_i$ does not attend to raw tokens outside this chunk. This implementation allows re-use of key-values for chunk representations $f_\theta(\mathbf{c}_i)$ in decoder for computing logits of a future chunk $\mathbf{c}_j$, where $j > i$. This way of com-

puting logits is quadratic in sequence length, in fact it is a constant times better: $O(\frac{N^2}{C})$ vs. the $O(N^2)$ complexity of the dense transformer, allowing for a potentially faster pre-training (see Section 4 for a discussion).

**Fast and Efficient Generation.** Due to compression, CATs can throwaway past chunks of tokens, and only keep their compressed chunk representations in memory. This straightaway results in a big reduction of memory; the KV cache is slashed by a factor of $C$, even for a modest chunk size of 4 (see Figure 2). This slash in memory is accompanied by reduced memory accesses the decoder makes in CATs, which is the major bottleneck during generation. The decoder attends to atmost $N_c + C$ tokens during generation, reducing compute required in self-attention significantly.

Implementing generation is simpler than training and very similar to how it occurs for a dense transformer. In fact, a pure PyTorch implementation[3] for CATs is on-par with efficient architectures that utilize custom kernels. Given a sequence, CATs first compute representations for each chunk in parallel and use them to prefill the decoder's KV cache. Then generation proceeds chunk by chunk: each new chunk is decoded token by token in the decoder, and once a chunk is complete, the chunk is compressed and its representation is prefilled in the KV cache for the generation of the next chunk. This loop continues until the sequence is fully generated. The full implementation details along with a PyTorch style pseudo-code are in Sections C and E.3.

### 2.2. CATs allow one to scale model parameters without proportionally increasing inference costs

The decoder in CAT contains the majority of parameters (as we will see in Sec. 3) and dominates the cost during inference. However, because the decoder operates on a reduced sequence length, both compute and memory requirements are lower than they would be for a dense transformer processing the full sequence. This computational structure in CAT enables the use of more parameters in decoder while maintaining inference costs that are comparable to, or even lower than, those of the same smaller parameter dense transformer. This *decoupling* of parameter count from inference cost is analogous to MoEs (Shazeer et al., 2017).

## 3. Experiments

### 3.1. Evaluation Criteria

We first motivate what the right evaluation criteria should be for comparing different models. While both training and inference costs are significant in developing and eventually deploying a language model, inference costs dominate

---

[3]Our implementation is inspired from: github.com/meta-pytorch/gpt-fast

in the long run (Timbers, 2023; Jassy, 2023). Moreover, these inference costs are increasing rapidly as inference-time scaling, reasoning models and long-horizon agents are becoming prevalent (Guo et al., 2025). Hence, **what ultimately matters for real-world deployment today is the performance a model achieves at a given inference budget.** This inference cost-first perspective suggests that raw parameter count alone can be a misleading common ground when comparing architecturally diverse models (having heterogeneous components, parameters etc.) — what matters is how those components and/or raw parameters translate to computation and memory usage during inference. MoEs (Shazeer et al., 2017) exemplify this principle: despite being heavily overparameterized relative to dense transformers, they are compared by inference cost rather than total parameters (Dai et al., 2024). For an extended discussion, see Section 4.

### 3.2. Baselines and Training Setup

**Baselines:** The CAT architecture complements any sequence mixer whose decoding cost depends on sequence length — instantiating CAT's decoder with such a mixer reduces costs due to the compressed sequence. Consequently, the only strict competitors are sequence mixers with length-independent costs, such as pure linear attention. Nonetheless, we compare broadly with recent state-of-the-art architectures to show that CAT with simple dense attention is already highly competitive.

Our evaluations include: (i) attention-based baselines: standard dense transformer (Touvron et al., 2023) and sparse transformer (Child et al., 2019), (ii) Linear Transformers such as Mamba2 (Dao & Gu, 2024) and GatedDeltaNet (GDN) (Yang et al., 2025b), as well as (iii) Hybrid architectures with GDN and attention layers interleaved in some pre-specified ratio. By default, all models below are configured with $L = 12$ layers and $D = 1024$ hidden dimension; any deviations are explicitly stated below.

Further, going beyond the default hyperparameter settings for the baselines, we scale up each model type by adjusting their model configurations to yield models at varying inference costs. In short, (i) we add scaled up versions of linear attention by increasing the recurrent state size $2\times$ (i.e. GDN-$2\times$), (ii) change dense-linear attention ratio in hybrid architectures (i.e. GDN-H 1:1, GDN-H 1:4), (iii) change the attention from sliding window to global in hybrids (i.e. GDN-H 1:7 G), (iv) scale up the model dimension by $2\times$ (i.e. GDN-H 1:7 G 2D, Sparse-4/8), (v) and for completeness, reduce it by $2\times$ too (e.g. Dense D/2, GDN-H 1:1 D/2). Most model types has atleast $\geq 2$ model configurations to ensure fair and broader comparison. This process results in a total of 12 models, all having different architectural components (dense attention, linear attention, hybrids), with

a range of parameter counts from $\sim$250M to upto $\sim$820M, and **varying** inference costs.

**We compare these 12 models against a *single* CAT model.**

**CAT configuration:** For CAT, we use $L = 12$ layers (same as baselines), and a wider hidden size of $D_g = 2D = 2048$ for the decoder, that takes up the majority of the parameters. The compressor is small and uses $L = 3$ layers and hidden size of $D_f = D = 1024$. Depth of compressor does not have major effect (App. D). This makes the parameter count for CATs close to $\sim (820 + 150)$ M parameters (similar to some models included in our comparison). We train CAT simultaneously on chunk sizes $C = \{4, 8, 16, 32\}$. This yields a single model that can work with different chunk sizes at once, offering test-time control of inference costs.

**Training setup:** All models were trained on 15B tokens of FineWeb-Edu (Penedo et al., 2024) with a context length of 4K following (Behrouz et al., 2024; Yang et al., 2025b). We use the AdamW optimizer (Loshchilov & Hutter, 2017) with a peak learning rate of 8e-4, weight decay of 0.1, gradient clipping of 1.0, batch-size of 0.5M tokens, employing the GPT2 tokenizer.

App. E provides details for each model and training.

### 3.3. Results

**Long-context recall:** Table 1 reports results on real-world in-context recall tasks from (Arora et al., 2024a). We report results on SWDE and FDA, which have longer sequences among the datasets in the suite (others have an average length of $\leq 300$ tokens (Arora et al., 2024b)). App. B.8 shows evaluations on all datasets. Figure 1b reports performance (i.e. in-context recall) for a given inference cost — specifically total memory usage, since it is the major inference time bottleneck in increasingly memory-bound GPU workloads (Gholami et al., 2024). **CAT performs as good or better than all models across inference cost budgets, using a single model only.** We additionally provide how in-context recall trades-off with latency (or compute) for the same set of models in appendix Figure 5. Linear models (Mamba2, GDN) lag far behind dense attention, while GDN-Hybrid variants reduces the gap. CAT surpasses nearly all efficient baselines, benefiting from the gracefully growing memory. CAT outperforms even the dense transformer at these tasks (at moderate chunk sizes $= 4, 8$), while have lower inference costs ($1.4\times$ faster and $2.2\times$ memory efficient). More details about the task can be found in Section E.2.

Table 3 reports results on the needle-in-haystack task (NIAH-N) from the RULER benchmark (Hsieh et al., 2024), that is, retrieve a 7 token number from long-context. CATs outperform the efficient baselines as context length increases, and interestingly show slower degradation with

*Table 1.* Zero-shot performance on real-world in-context recall tasks measured upto 4K sequence lengths. H and G stand for Hybrid and Global respectively. Fig. 1b gives an inference costs (memory) matched comparison. All CATs here are a single model.

| Model | SWDE | FDA | Avg. ↑ |
|---|---|---|---|
| Dense | 43.4 | 19.7 | 32.0 |
| Dense D/2 | 32.0 | 22.0 | 27.0 |
| Sparse-4 | 36.0 | 16.0 | 26.0 |
| Sparse-8 | 20.9 | 6.0 | 13.0 |
| Mamba2 | 13.5 | 4.5 | 9.0 |
| GDN | 18.0 | 6.8 | 12.0 |
| GDN-2× | 24.0 | 11.0 | 17.5 |
| GDN-H 1:1 | 44.0 | 17.8 | 31.0 |
| GDN-H 1:1 D/2 | 17.0 | 3.2 | 10.0 |
| GDN-H 1:4 | 33.0 | 20.0 | 27.0 |
| GDN-H 1:7 G | 38.0 | 25.0 | 32.0 |
| GDN-H 1:7 G 2D | 46.0 | 36.0 | 41.0 |
| CAT-4 | **49.1** | **45.1** | **47.1** |
| CAT-8 | 38.2 | 34.8 | 36.5 |
| CAT-16 | 27.5 | 15.4 | 21.5 |
| CAT-32 | 13.2 | 3.2 | 8.2 |

length, even compared to the dense transformer. This slow degradation can possibly be attributed to reduced sequence length in CAT that leads to fewer *distractions* for attention (Barbero et al., 2024; Vasylenko et al., 2025; Chiang & Cholak, 2022; Golovneva et al., 2025). More discussion can be found in App. B.7, where we extend these results to the harder task from RULER (namely NIAH-U, that is retrieve 32 token `uuids`).

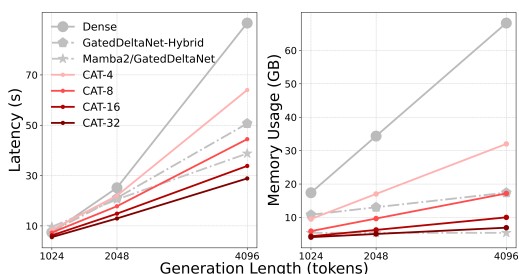

Figure 2. A single CAT model generates $1.4 - 3.2\times$ faster than the dense transformer while showcasing upto $2.2 - 9.5\times$ lower memory usage.

**Long-context language modeling and understanding:** We test language modeling and understanding on long contexts (upto 4K contexts). As standard perplexity (or test log-loss) averaged over all tokens **does not** indicate downstream long-context ability (Fang et al., 2025; Liu et al., 2023; Hu et al., 2024): Table 2 conducts evaluations on the LongPPL (Fang et al., 2025) metric, that calculates loss on a few key tokens which are essential for long-context

understanding. We employ Llama-3.1-8B as evaluator. For completeness, we report test log-loss averaged over all tokens in App. B.2. Additionally, Table 2 reports evaluation on LongBench (Bai et al., 2023) that tests for long-context understanding. All CATs perform competitively.

*Table 2.* LongPPL (Fang et al., 2025) (we report `log_loss`) and zero-shot evaluation on a suite of tasks from LongBench (Bai et al., 2023); upto 4K tokens. Refer to App. B.2 for standard perplexity, Table 5 for task-wise results on LongBench, and to Section E.2 for details. All CATs are a single model.

| Model | LongPPL ↓ | | LongBench ↑ |
|---|---|---|---|
| | GovReport | PG19 | Avg. |
| Dense | 4.50 | 5.54 | 9.3 |
| Sparse-8 | 5.09 | 5.54 | 9.3 |
| Mamba2 | 4.71 | 5.21 | 8.0 |
| GDN | 4.59 | 5.02 | 8.9 |
| GDN-2× | 4.23 | 4.86 | 8.1 |
| GDN-H 1:1 | 4.55 | 5.33 | 9.0 |
| GDN-H 1:4 | 4.49 | 5.04 | 11.6 |
| GDN-H 1:7 G | 4.31 | 4.94 | 12.2 |
| GDN-H 1:7 G 2D | 4.02 | 5.04 | 11.6 |
| CAT-4 | **2.96** | **4.20** | **13.9** |
| CAT-8 | 3.19 | 4.30 | 12.1 |
| CAT-16 | 3.66 | 4.57 | 9.5 |
| CAT-32 | 4.36 | 4.92 | 7.9 |

**Short-context language modeling and understanding benchmarks:** Table 7 in appendix reports zero-shot accuracies on key common-sense reasoning benchmarks; All CAT variants outperform existing efficient baselines except one. However, note that these evaluations only consider short sequences ($\leq$ 30 tokens on average).

**Benchmarking generation:** Figure 2 compares architectures as one scales the sequence length, with a fixed batch-size of 320 to maximize throughput. CAT generates sequences $1.4 - 3.2\times$ **faster** than the dense transformer while showcasing **upto** $2.2 - 9.5\times$ **lower total memory usage** as one increases chunk sizes, despite using significantly more parameters than the baselines due to wider decoder and the additional compressor. This is not surprising since the major bottlenecks during generation are: (a) KV cache size that drives the main memory requirement during generation and not the parameter count (Sec. 4), (b) memory accesses required for a token, and (c) FLOPs used per token determined by the past tokens being attended to. CATs reduce these factors despite carrying more parameters overall. App. E.3 provides implementation details.

**Appendix:** We provide additional results demonstrating CAT's flexibility and scalability. First, we show CAT can be used as a drop-in layer (App. B.10) and that its compressor and decoder can make use of any sequence mixer (App. B.11). We then present scaling experiments (App. 6), ablations on design choices (App. D), evaluations on the

synthetic MQAR task (App. B.5 – tested on $4\times$ longer sequences than usual), and analysis of perplexity across chunk boundaries (App. B.14).

*Table 3.* Accuracy on RULER (Hsieh et al., 2024) S-NIAH-N benchmark. All CATs are a single model.

| | S-NIAH-N (↑) | | |
|---|---|---|---|
| **Model** | **1K** | **2K** | **4K** |
| Dense | 96.0 | 92.0 | 43.0 |
| Sparse | 51.2 | 46.2 | 5.0 |
| Mamba2 | 97.7 | 81.1 | 18.6 |
| GDN | 84.7 | 69.1 | 13.6 |
| GDN-2× | 78.0 | 61.4 | 29.0 |
| GDN-H 1:1 | 99.0 | 97.0 | 44.0 |
| GDN-H 1:4 | 98.0 | 96.0 | 35.8 |
| GDN-H 1:7 G | 98.3 | 93.0 | 23.2 |
| GDN-H 1:7 G 2D | **99.5** | **99.3** | 43.8 |
| CAT-4 | 96.0 | 97.0 | **96.0** |
| CAT-8 | 90.0 | 93.0 | 91.0 |
| CAT-16 | 76.0 | 72.0 | 70.0 |
| CAT-32 | 60.0 | 37.0 | 31.0 |

## 4. Discussion

**Is it fair that CATs use more parameters?** In the main results, CATs use more parameters than some models in our comparison, while still using similar or lower inference costs. We acknowledge that increased parameters in CAT may contribute to its strong performance. That being said, we argue this slower growth of inference cost as a function of parameters is a feature that CAT provides because of compression. Most sequence mixers & architectures are *monolithic* and incur relatively higher inference costs when increasing parameters. Analogous to MoEs, CATs utilize additional parameters *cleverly* through their compressed computational structure without proportionally blowing costs.[4] This is particularly advantageous in modern long-context scenarios, where cache memory dominates over parameter counts. Further, CAT does outperform models that have similar parameter counts at matched or lower inference costs (i.e. Sparse-4/8 and GDN-H 1:7 G 2D).

From a deployment perspective, what matters is performance at a given inference cost: if CATs can deliver better performance at the same cost, perhaps owing to their additional parameters, this is a desirable property. To motivate this further, we ask a question: *suppose model A has more parameters than model B, then if model A outperforms model B using lower inference costs, does it matter that model A has more parameters than model B? Which model should one deploy: model A or B?*

---

[4]Note that while MoEs do enable scaling of parameters while controlling costs, they are not sequence mixers since they operate on the feedforward layer and can be applied to any mixer, including CATs.

Nevertheless, for completeness, we provide results when CAT is parameter-matched to lower parameter baselines in Section B.4.

Empirically, we observed that wider decoder in CAT substantially improves performance when decoding from compressed chunk representations (App. B.4 and D.2). This need for additional capacity, specifically higher dimensionality, to enable accurate decoding from compressed inputs has been noted in prior work (Li et al., 2024a; Ho et al., 2024; Yu et al., 2023), and may reflect a fundamental interaction between available information and the compute required to process it (Xu et al., 2020).

**What's the training cost of CATs?** While CATs reduce attention FLOPs significantly due to compressed sequence lengths, the overall training cost depends on both attention and feedforward layers. At 4K sequences, feedforward FLOPs dominate over attention (Austin et al., 2025), resulting in $\sim 2.5\times$ higher training FLOPs for CATs. However, despite more parameters, this overhead reduces to just $\sim 1.25\times$ at 16K sequences as attention becomes the bottleneck. More importantly, **at any sequence length**, CATs **amortize** training cost by training a single unified model that exposes multiple inference costs. Training multiple independent models from scratch that target exactly those inference costs would take more FLOPs overall. Moreover, this gap to train a single CAT versus multiple independent models widens as pretraining moves to longer sequences. Further, as inference costs are starting to dominate the total cost of models, the ability to serve multiple efficiency points from a single model becomes increasingly valuable compared to training costs. App C.5 provides more details and discussion.

**Are CATs adoptable?** Inference costs dominate in the long run, and it is increasingly memory-bound rather than compute-bound (Gholami et al., 2024). For instance, Qwen3-14B at a modest batch size of 16 requires an order of magnitude more memory for the KV cache than for the model weights: 28GB for weights versus ∼670GB for the KV cache at maximum context length. A CAT variant of the same model could reduce total memory usage by up to ∼4× despite having more parameters overall, enabling longer sequences on the same hardware and higher generation throughput.[5] These memory reductions become even more pronounced at larger batch sizes, that power important industrial workloads such as synthetic data generation (Maini et al., 2025) and large-scale rollouts in post-training pipelines (Noukhovitch et al., 2024; Zhang & Ranganath, 2025).

---

[5]Total memory usage for CATs: $28 \cdot \left(4 + \frac{1}{4}\right) + \frac{670 \cdot 2}{32} \approx 160$GB at chunk size $C = 32$.

## 5. Related Work

We summarize the most relevant related work here. Importantly, CAT is a meta-sequence mixer: most approaches described below can be used within, or mixed with CAT. **Table 4 in appendix** highlights these relationships and conceptual differences. App. F provides extended discussion.

**Efficient sequence mixers:** Sparse or sliding window attention (Child et al., 2019; Zaheer et al., 2020; Jiang et al., 2023) heuristically restrict which tokens are attended to. This reduces compute (and sometimes memory), but if the wrong mask is chosen, these methods underperform or require more depth (Arora et al., 2024a). Matching dense transformer quality often requires large windows or composition with dense attention at specific layers (Arora et al., 2024a; Agarwal et al., 2025). Linear attention (Katharopoulos et al., 2020; Arora et al., 2024a; Dao & Gu, 2024; Yang et al., 2025b) replaces softmax with kernelized attention, admitting a recurrent form with constant memory. Recent variants add data-dependent gating (Dao & Gu, 2024; Yang et al., 2025b), but all require handcrafted state update rules. The fixed-size recurrent state struggles with long-range recall (Arora et al., 2024a; Jelassi et al., 2024; Wen et al., 2024), and making these mixers competitive requires careful composition with attention – a process that involves significant trial-and-error (Waleffe et al., 2024; Qwen, 2025; Wang et al., 2025). CAT complements any sequence mixer whose decoding cost depends on sequence length, since instantiating CAT's decoder with such a mixer reduces costs in that mixer due to the compressed sequence. Consequently, the only strict competitors are sequence mixers with length-independent costs, such as pure linear attention. Otherwise, any existing efficient sequence mixer can serve in CAT, or be used alongside CAT for test-time flexibility, including hybrid designs (App. B.11).

Mixture-of-Experts (MoEs) (Shazeer et al., 2017; Dai et al., 2024) take a different approach: they increase parameters in feed-forward layers via sparse computation, without increasing inference costs – making them complementary to any sequence mixer, including those that CAT utilizes in compressor and decoder.

**Compressing past context:** Recurrent compression (Rae et al., 2020; Chevalier et al., 2023) enables generation of longer sequences on limited compute and memory. However, sequential training is slow and memory-intensive, scaling poorly on modern hardware that favors parallelism. Training recurrent models also poses optimization challenges; Geiping et al. (2025) required careful recipes to prevent collapse when scaling up. Native Sparse Attention (NSA) (Yuan et al., 2025) attends to compressed chunks as well as few raw tokens, with compression happening in parallel at every layer. This is similar in spirit to CAT, but NSA retains the full KV cache for the entire context—yielding compute savings but no memory savings during inference.

**Hierarchical transformers:** Hourglass architectures (Nawrot et al., 2021; 2022; Slagle, 2024) downsample the sequence into coarse tokens, then upsample before decoding. This saves compute during training, but generation still requires memory accesses for all past tokens, especially *fine-grained* ones, which is the main bottleneck. MegaByte and Block Transformer (Ho et al., 2024; Yu et al., 2023) model sequences as independent chunks conditioned on a single compressed representation of the past. While this improves efficiency, the fixed-size bottleneck hurts recall even on simple tasks (see App. B.6). In contrast, CAT's decoder attends to *all* past chunk representations, enabling flexible memory that grows gracefully with sequence length while maintaining efficiency.

In summary, CAT complements most existing approaches, can extend them, or be mixed with them to unlock test-time control of inference cost.

## 6. Conclusion and Future Work

This paper posed two research questions initially: (i) can we develop an approach where per-token inference cost is controllable at test time? and (ii) can we instantiate such an approach using simple, standard components that perform comparably to specialized approaches at similar inference costs? We answer both affirmatively: CAT allows for test-time control of per-token inference costs, and when instantiated with simple dense attention produces a single CAT model that is as powerful or better than many popular efficient baselines at similar inference costs – without requiring any specialized machinery or careful design choices. More broadly, we emphasize that the value of CAT lies not in any single benchmark comparison, but in the flexibility it provides to control inference costs – making it a practical and compelling choice for real-world deployment.

**Future work:** One interesting direction is data-dependent adaptivity. CATs, as they stand, require users to choose a chunk size appropriate for their compute and memory budgets. Instead, one could post-train to allow CATs to learn to allocate budget themselves based on the context and task. Such post-training would enable truly adaptive efficiency. Another possibility is using CAT recursively inside CAT. Finally, scaling up CATs to larger model scales is an interesting direction.

**Impact statement.** This work shows a new approach to build efficient language models. There are many potential societal consequences of this work as a consequence of focusing on efficient language models, but none specific to the work itself.

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

# A. Related Work Table

| Method | Unrestricted Access to Memory? | Flexible memory? | Scalable training? | Both compute & memory efficient? | Adaptive? | Mixable with CAT? | Usable in CAT's compressor/decoder? |
|---|---|---|---|---|---|---|---|
| **Dense**: (Vaswani et al., 2017) | ✓ | ✓ | ✓ | ✗ | ✗ | ✓ | ✓ |
| **Sparse Attention**: (Child et al., 2019) | ✗ | ✓ | ✓ | ✓ | ✗ | ✓ | ✓ |
| **NSA**: (Yuan et al., 2025) | ✓ | ✓ | ✓ | ✗ | ✗ | ✓ | ✓ |
| **Sliding window Attn.**: (Jiang et al., 2023) | ✗ | ✗ | ✓ | ✓ | ✗ | ✓ | ✓ |
| **Linear Attention**: (Dao & Gu, 2024) | ✓ | ✗ | ✓ | ✓ | ✗ | ✓ | ✓ |
| **Recursive compression**: (Chevalier et al., 2023) | ✓ | ✓ | ✗ | ✓ | ✗ | ✓ | ✓ |
| **MegaByte/Block Transformer**: (Ho et al., 2024; Yu et al., 2023) | ✓ | ✗ | ✓ | ✓ | ✗ | ✓ | ✓ |
| **CATs** | ✓ | ✓ | ✓ | ✓ | ✓ | ✓ | ✓ |

*Table 4.* We categorize existing related work by key properties desirable for an efficient architecture, and indicate whether CAT can complement these approaches as a **meta-sequence mixer**. *"Both compute and memory efficient?"* signifies savings during inference; *"Unrestricted Access to Memory"* signifies whether an architecture can freely access any part of the memory in the past, without any artificial restrictions; *"Mixable with CAT?"* indicates whether CAT as a layer be used in these approaches; *"Usable CAT's compressor/decoder?"* indicates whether the method can serve as a compressor or decoder within CAT. Note that CAT itself can be recursively used as compressor/decoder.

# B. More experiments

## B.1. LongBench

| | Single-doc QA | | Multi-doc QA | | Few Shot | | Avg. |
|---|---|---|---|---|---|---|---|
| **Model** | QAS | MQA | HQA | 2WMQ | TQA | TREC | |
| Dense | 3.9 | 12.2 | 6.9 | **10.8** | 11.2 | 10.6 | 9.3 |
| Sparse-8 | 5.1 | 11.0 | 7.0 | 10.6 | 10.5 | 5.6 | 9.3 |
| Mamba2 | 4.1 | 11.9 | **7.6** | 7.6 | 9.0 | 7.6 | 8.0 |
| GDN | **8.3** | **15.5** | 6.0 | 7.9 | 7.4 | 8.3 | 8.9 |
| GDN-2× | 4.1 | 11.8 | 6.7 | 9.6 | 9.8 | 6.8 | 8.1 |
| GDN-H 1:1 | 4.2 | 13.3 | 6.6 | 11.6 | 11.8 | 6.5 | 9.0 |
| GDN-H 1:4 | 4.6 | 13.0 | 7.0 | 10.4 | 10.6 | 24.2 | 11.6 |
| GDN-H 1:7 G | 4.7 | 12.5 | 6.4 | 12.2 | 9.2 | 28.3 | 12.2 |
| GDN-H 1:7 G 2D | 3.9 | 12.9 | 8.3 | 9.1 | 11.9 | 23.7 | 11.6 |
| CAT-4 | 5.6 | 12.7 | 7.4 | 9.9 | 12.1 | **35.6** | **13.9** |
| CAT-8 | 5.5 | 11.0 | 6.1 | 8.0 | **12.4** | 29.5 | 12.1 |
| CAT-16 | 4.3 | 14.1 | 6.1 | 5.6 | 10.5 | 16.6 | 9.5 |
| CAT-32 | 4.7 | 11.0 | 7.0 | 6.6 | 10.0 | 8.3 | 7.9 |

Table 5. Zero-shot evaluation of baselines on suite of tasks from LongBench (Bai et al., 2023) up to 4K tokens. Refer to Section E.2. CAT-4/8/16/32 are a single model.

## B.2. Standard Perplexity averaged over all tokens

We report standard perplexity metric here — that averages over all tokens in the context. Note that this metric is an unreliable proxy for long-context performance (Fang et al., 2025). We report upto 4K tokens.

| | PPL ↓ | |
|---|---|---|
| **Model** | GovReport | PG19 |
| Dense | 2.61 | 3.19 |
| Sparse | 2.61 | 3.15 |
| Mamba2 | 2.63 | 3.18 |
| GDN | 2.59 | 3.13 |
| GDN-2× | 2.57 | 3.13 |
| GDN-H 1:1 | 2.56 | 3.14 |
| GDN-H 1:4 | 2.56 | 3.16 |
| GDN-H 1:7 G | 2.56 | 3.12 |
| GDN-H 1:7 G 2D | 2.39 | 2.93 |
| CAT-4 | 2.55 | 3.14 |
| CAT-8 | 2.56 | 3.14 |
| CAT-16 | 2.60 | 3.16 |
| CAT-32 | 2.67 | 3.20 |

Table 6. We report test log-loss upto 4K tokens. CAT-4/8/16/32 are a single model.

## B.3. Short-context language understanding evaluations

| Model | HS↑ | PQ↑ | AE↑ | AC↑ | WG↑ | OQA↑ | Avg.↑ |
|---|---|---|---|---|---|---|---|
| Dense | 34.8 | 65.6 | 56.7 | 24.4 | 51.1 | 20.0 | 42.1 |
| Sparse-8 | 35.6 | 66.8 | 57.3 | 25.4 | 51.1 | 22.8 | 43.2 |
| Mamba2 | 36.1 | 67.0 | 59.2 | 26.5 | 51.9 | 21.6 | 43.7 |
| GDN | 36.1 | 66.8 | 58.7 | 25.2 | 51.6 | 22.8 | 43.5 |
| GDN-2× | 35.9 | 67.4 | 58.6 | 27.2 | 51.8 | 21.8 | 43.8 |
| GDN-H 1:1 | 36.8 | 66.3 | 56.4 | 25.8 | 52.1 | 20.4 | 43.0 |
| GDN-H 1:4 | 34.8 | 67.0 | 57.0 | 26.5 | 50.3 | 22.0 | 42.9 |
| GDN-H 1:7 G | 36.0 | 67.6 | 57.4 | 26.6 | 51.5 | 23.4 | 43.7 |
| GDN-H 1:7 G 2D | 38.6 | 70.1 | 62.3 | 27.8 | 52.7 | 23.6 | 45.8 |
| CAT-4 | 35.6 | 66.4 | 59.5 | 27.1 | 51.5 | 23.4 | 43.9 |
| CAT-8 | 35.4 | 66.8 | 60.1 | 27.4 | 51.3 | 23.6 | 44.1 |
| CAT-16 | 35.5 | 67.3 | 60.2 | 27.0 | 52.0 | 23.8 | 44.3 |
| CAT-32 | 35.9 | 68.2 | 61.0 | 27.0 | 53.6 | 25.0 | 45.1 |

Table 7. Zero-shot accuracy on common-sense reasoning benchmarks. CAT outperforms all models on common-sense reasoning evaluations except one. However, note that these evaluations considers short sequences only (≤ 30 tokens on average). Hence, we test language understanding on longer contexts in Table 2 on LongBench (Bai et al., 2023), LongPPL (Fang et al., 2025) and test in-context recall on real world tasks (Arora et al., 2023b) in the main text. Section E.2 expands the acronyms in Table 7.

## B.4. Parameter-Matched Results

Here, we report results for parameter-matched comparisons to complement our inference-cost-based analysis in the main paper. We acknowledge that both parameter count and inference cost are axes for model comparison. That being said, we believe inference-cost is the correct comparison criterion from a deployment point of view.

To provide parameter-matched comparison, we train a lower parameter CAT variant. Results are shown in Table 8. At matched parameters, CAT-4 achieves competitive performance on LongPPL (GovReport: 3.70, PG19: 4.71) and retrieval tasks (Avg Recall: 26.0), while providing memory reduction (4.2×) and generation speedup (2.9×) over dense transformer. Notably, CAT-4 outperforms parameter-matched alternatives like Mamba2 (Avg Recall: 9.0) and GDN (Avg Recall: 12.0) on recall tasks, demonstrating that CAT's benefits may not be solely attributable to increased parameter count.

**Further, note that CAT with a wider decoder (reported in the main text) outperforms baselines while still providing lower inference costs despite using more parameters, making it very desirable for deployment. Not to mention CAT provides controllable inference-costs.**

*Table 8.* We report evaluations when CATs are parameter matched with baselines. Table includes memory reduction and generation speed-ups with respect to a dense transformer. GovReport and PG19 columns report LongPPL metric (using log-loss). LongBench reports average across all tasks. We additionally include results for CAT with a wider decoder.

| Model | Params | Mem. Red. ↑ | Gen. Speedup ↑ | GovReport ↓ | PG19 ↓ | LongBench ↑ | SWDE ↑ | FDA ↑ | Avg Recall ↑ |
|---|---|---|---|---|---|---|---|---|---|
| Dense | 260M | 1.0× | 1.0× | 4.50 | 5.54 | 9.3 | 43.4 | 19.7 | 32.0 |
| Mamba2 | 260M | 6.4× | 2.4× | 4.71 | 5.21 | 8.0 | 13.5 | 4.5 | 9.0 |
| GDN | 310M | 6.4× | 2.4× | 4.59 | 5.02 | 8.9 | 18.0 | 6.8 | 12.0 |
| GDN-2× | 310M | 3.5× | 1.2× | 4.23 | 4.86 | 8.1 | 24.0 | 11.0 | 17.5 |
| GDN-H 1:1 | 280M | 2.0× | 1.8× | 4.55 | 5.33 | 9.0 | 44.0 | 17.8 | 31.0 |
| GDN-H 1:4 | 280M | 3.3× | 2.4× | 4.31 | 4.94 | 11.6 | 33.0 | 20.0 | 27.0 |
| Parameter matched CATs | | | | | | | | | |
| CAT-4 | 75M + 260M | 4.2× | 2.9× | 3.70 | 4.71 | 7.5 | 29.2 | 22.8 | 26.0 |
| CAT-8 | 75M + 260M | 7.6× | 4.4× | 4.00 | 4.83 | 7.8 | 21.8 | 15.9 | 18.8 |
| CAT-16 | 75M + 260M | 12.2× | 5.8× | 4.57 | 5.11 | 6.0 | 11.0 | 4.8 | 7.9 |
| CAT-32 | 75M + 260M | 15.7× | 7.1× | 5.10 | 5.52 | 5.6 | 8.2 | 4.0 | 6.1 |
| CATs with wider decoder (reported in the main text) | | | | | | | | | |
| CAT-4 | 150M + 820M | 2.0× | 1.4× | **2.96** | **4.20** | **13.9** | **49.1** | **45.1** | **47.1** |
| CAT-8 | 150M + 820M | 3.5× | 2.0× | 3.19 | 4.30 | 12.1 | 38.2 | 34.8 | 36.5 |
| CAT-16 | 150M + 820M | 5.5× | 2.7× | 3.66 | 4.57 | 9.5 | 27.5 | 15.4 | 21.5 |
| CAT-32 | 150M + 820M | 6.8× | 3.2× | 4.36 | 4.92 | 7.9 | 13.2 | 3.2 | 8.2 |

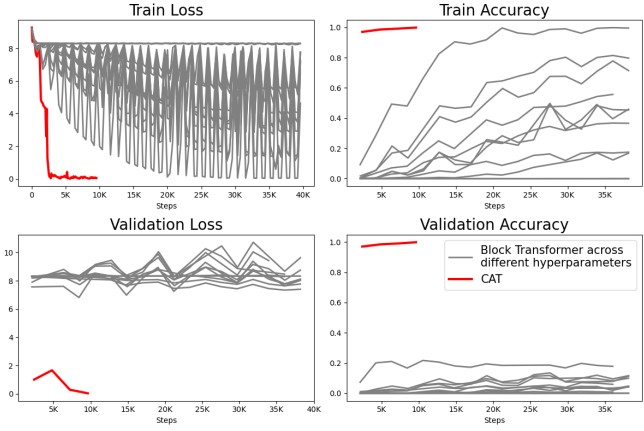

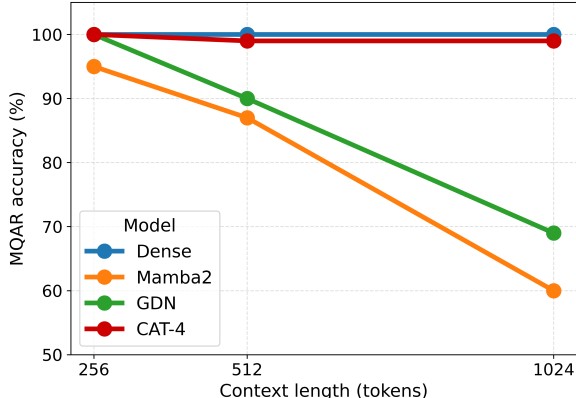

*Figure 3.* Block Transformer (Ho et al., 2024; Yu et al., 2023) (across different configurations and hyperparameters) fails to solve a simple MQAR task with only 4 key-value pairs tested on modest sequence length of 256 tokens. Note that training of CAT stops when it solves the task perfectly.

*Figure 4.* Comparison of different architectures across sequence lengths on MQAR task. We measure test-accuracy on the hardest subset. All architectures are memory matched in bytes at every point (except dense transformer).

### B.5. CATs outperform baselines when memory matched

To rule out any memory discrepancy, (Fig. 2) evaluates on MQAR task (Arora et al., 2023a), matching memory budgets down to the level of bytes, and stress-tests models up to 1K sequence length (5× standard); Figure 4 in reports results. Baselines are grid-searched over learning rates. Linear models collapse at longer contexts, while CATs remain near-perfect, thanks to the flexible yet efficient memory scaling. We use the same setup in App. B.9.

### B.6. Comparison with MegaByte/Block Transformer

The MegaByte/Block Transformer (Ho et al., 2024; Yu et al., 2023) has elements similar to CAT but fail to solve a simple in-context recall task in Figure 3 across different hyperparameters and architecture configurations due to the fixed memory bottleneck. In fact, the block transformer overfits on the task. CATs alleviate the memory bottleneck with a gracefully growing memory, allowing it to solve the task, with even lower memory requirements.

In figure 3, we evaluate in-context recall ability for Block Transformer architectures (Ho et al., 2024; Yu et al., 2023), that model chunks of tokens similar to CATs but with a subtle but salient difference in the architecture circuit (that we explain below). For this experiment, we test on the MQAR task (a synthetic needle-in-haystack task (Arora et al., 2023a)) on a modest sequence length of 256. We test the accuracy of retrieving just 4 needles. We parametrize components of Block Transformer that is: global model and local model using a transformer, the embedder is a look-up table or a transformer. We keep the patch size/chunk size as 4 – same as CAT. We keep the identical training setup for both architectures. We grid search for hyper-parameters (lr, hidden_size, and embedder parameterization), even **using more memory** than the CAT baseline, in its global decoder. Even in these simple settings and added advantage, Block Transformer (Ho et al., 2024; Yu et al., 2023) fails to solve the task (fig. 3) – instead the model starts to memorize the train points, as seen from train loss and train accuracy – train metrics keep getting better, however, test metrics suffer.

CATs directly pass all the "local" patch/chunk representations directly to the decoder, unlike the block transformer that forces the history to be compressed into fixed dimensional representation. This design choice helps CAT *alleviate the memory bottleneck* that (Ho et al., 2024) suffers from where the architecture must compress everything from the past into a single "global" representation to generate the next chunk. Note that this different design choice in CATs does not introduce any memory/compute overhead compared to Block Transformer (Ho et al., 2024), it just changes the circuit of the architecture. In fact, CATs don't utilize three different components (embedder, global decoder, local decoder) – it only uses a compressor and a decoder, reducing the design space and (significant) parameter requirements further.

### B.7. RULER benchmark

Table 3 (in the main text) reported results on RULER (Hsieh et al., 2024) single-needle tasks: S-NIAH-N (recall number from the context). We observed linear recurrent models (Mamba2, GDN) struggle at longer contexts, and while GDN-Hybrid narrows the gap with dense transformers, performance still drops at longer contexts. CATs-4/8/16 outperform the efficient baselines as context length increases, showing slower degradation with length, even compared to the dense transformer. This slow degradation can possibly be attributed to reduced sequence length in CAT that leads to fewer *distractions* for attention (Barbero et al., 2024; Vasylenko et al., 2025; Chiang & Cholak, 2022; Golovneva et al., 2025). Further, large-chunk CAT underperforms at short contexts but interestingly surpass baselines at long ones. One reason why large-chunk CAT underperforms could be due to ineffective compression – due to larger chunks, the compressor in CAT is not always able to surface the *right* information in the chunk representation for accurate retrieval. More pre-training or finetuning on specific task data alleviates this problem for large chunk CAT (see App. B.15). That being said, there is an upper limit to how much information fixed sized chunk representations can practically learn to hold for large token chunks.

Table 9 further reports results on the harder S-NIAH-U (recall a long alpha-numeric string or UUID).

*Table 9.* Accuracy on RULER (Hsieh et al., 2024) S-NIAH-U benchmark.

| | S-NIAH-U | | |
|---|---|---|---|
| **Model** | **1K** | **2K** | **4K** |
| Dense | **93.6** | 55.7 | 19.8 |
| Sparse | 12.8 | 1.4 | 0.8 |
| Mamba2 | 46.7 | 4.6 | 1.0 |
| GDN | 38.9 | 2.6 | 2.0 |
| GDN-H 1:1 | 50.9 | 5.6 | 2.6 |
| CAT-4 | 79.6 | **59.3** | 46.5 |
| CAT-8 | 68.1 | 57.5 | **47.3** |
| CAT-16 | 10.0 | 6.6 | 3.8 |
| CAT-32 | 0.0 | 0.0 | 0.0 |

### B.8. Recall evaluation

Here, we evaluate all baselines on all datasets from the EVAPORATE suite of tasks that tests for real-world in-context recall.

| Model | SWDE | FDA | Squad | TriviaQA | Drop | Avg. |
|---|---|---|---|---|---|---|
| Dense | 43.4 | 19.7 | 31.0 | 15.0 | 19.4 | 26.7 |
| Sparse | 20.9 | 6.0 | 20.7 | 15.2 | 19.3 | 16.4 |
| Mamba2 | 13.5 | 4.5 | 24.9 | 13.9 | 17.8 | 14.9 |
| GDN | 18.0 | 6.8 | 25.5 | **15.5** | 17.2 | 16.6 |
| GDN-H 1:1 | 44.0 | 17.8 | **32.9** | 15.4 | **19.8** | 26.0 |
| CAT-4 | **49.1** | **45.1** | 28.3 | 15.0 | 17.9 | **31.1** |
| CAT-8 | 38.2 | 34.8 | 25.9 | 14.0 | 18.3 | 26.2 |
| CAT-16 | 27.5 | 15.4 | 20.4 | 14.8 | 16.9 | 18.9 |
| CAT-32 | 13.2 | 3.2 | 15.8 | 13.0 | 14.3 | 11.9 |

*Table 10.* Zero-shot performance on real-world in-context recall tasks from EVAPORATE suite, measured upto 4K sequence lengths. Note that only SWDE and FDA have long token sequences among the datasets in the suite (others have an average length of $\leq 300$ tokens (Arora et al., 2024b)).

## B.9. Sparse or Sliding Window Attention needs more layers for recall

We evaluate models on the synthetic multi-associate query recall (MQAR) task, proposed in (Arora et al., 2023a) and further popularized in (Arora et al., 2024a). All models use depth of 2 layers, and are trained and tested on sequence lengths upto 256 having varying number of key-value pairs. CAT models use a 1 layer compressor, followed by a 2 layer decoder, with a chunk size of 4, both using model dimension of $D = D_d = 64$ in this case. Note that the state size for CAT is $\frac{N}{C} \cdot D = 4096$ for this particular sequence length and model dimension. Sparse attention uses a chunk size of 4 (for fair comparison with CAT); Sliding window uses a window size of 64.

| Method | Solves? | State Size |
|---|---|---|
| Dense | ✓ | 16384 |
| Sparse | ✗ | 4096 |
| Sliding Window | ✗ | 4096 |
| CAT | ✓ | 4096 |

*Table 11.* For each method, we report the state size at which the particular method was trained for the MQAR task. Each method was grid searched for best possible hyper-parameters. We use the state size calculations provided in (Arora et al., 2024a; 2023a).

In table 11, CAT is able to solve the MQAR task. Notably, we find the sparse attention as well as sliding window attention fail to solve the task at 2 layers, highlighting their dependence on depth.

### B.10. CAT as a layer

While CAT presented in the paper is a separate meta-sequence mixer, one can take the core concepts and instantiate CAT as a layer that can be swapped in any sequence model as a drop-in replacement. This can unlock lots of interesting possibilities starting with creating hybrid as well as adaptive architectures that mixes CAT layers alongside dense attention, or perhaps even linear attention. We leave this open for future work.

To instantiate CAT as a seperate layer in itself, we parameterize the *compressor* as a simple linear projection. We use the dense attention mechanism itself as the *decoder*. Before applying the compression and decoding from compressed chunk representations, we artificially up-project the input embeddings in the layer – this is done following the observation in the main paper that decoding from compressed representations requires higher dimensionality. We will release the implementation in our code. Table 13 reports MQAR accuracy when CAT is used as a layer. We use a fixed chunk size of 4 in this experiment. We use 2 layers of CAT. Rest of the setup follows (Arora et al., 2024a).

| Method | Solves? | State Size |
|---|---|---|
| Dense | ✓ | 16384 |
| CAT | ✓ | 4096 |
| CAT (layer) | ✓ | 4096 |

*Table 12.* CAT instantiated as a seperate layer solves the MQAR task.

### B.11. CAT is a meta sequence mixer

CAT has two components: a compressor and a decoder – each of these could make use of any sequence mixers, such as linear attention. Here we provide preliminary result on the MQAR task where the decoder in CAT is a GDN-HYBRID architecture (Yang et al., 2025b) (having a 1:1 dense-to-linear ratio). This new architecture solves this task, empirically demonstrating the use of linear attention layers inside of CAT: meaning rather than CAT being a strict competitor to linear attention (or any other efficient sequence mixer), **CAT is complementary**. Further, the use of a different sequence mixers inside of CAT can unlock the test-time control of efficiency with those sequence mixers (e.g., GDN-HYBRID in this case).

CAT solves the MQAR task when the decoder is instantiated as a GDN-Hybrid architecture (1:1 dense-linear ratio) with 2 layers. We use the same setup described in (Arora et al., 2024a): using sequences upto 256 with maximum key-values in the sequence. The chunk size used is set to 4.

| Method | Solves? |
|---|---|
| Dense | ✓ |
| CAT | ✓ |
| CAT (GDN-HYBRID decoder) | ✓ |

*Table 13.* The decoder in CAT is replaced with a GDN-HYBRID architecture. The resulting CAT architecture solves the MQAR task.

## B.12. Quality and latency trade-off

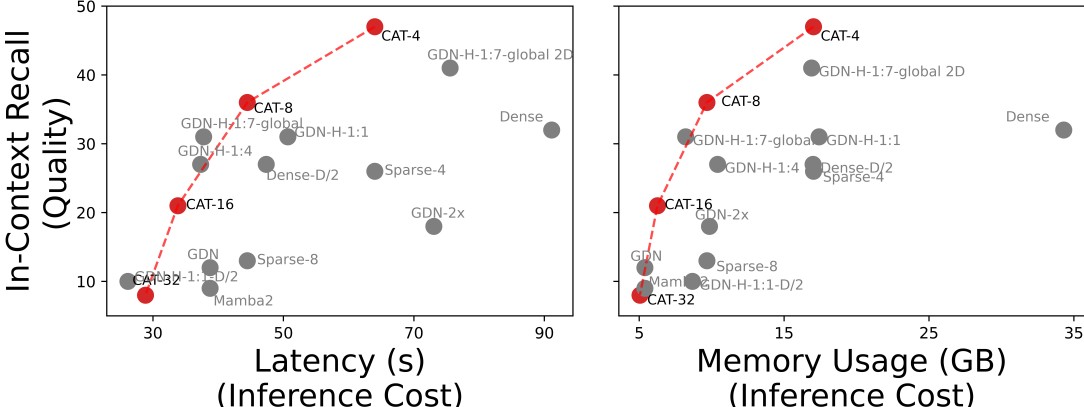

*Figure 5.* In the main text, we compared different models by measuring what quality (in-context recall) they get for a given inference cost, specifically memory usage, since memory usage is the major inference time bottleneck in increasingly memory-bound GPU workloads (Gholami et al., 2024). We report quality and latency trade-off here for completion. CAT achieves a pareto-frontier in quality (in-context recall) and inference costs (memory usage) trade-off curve across 12 models. At the same time, CAT outperforms most models given a desired latency requirement. CAT achieves this using a *single* model only.

## B.13. CATs scale as well as their dense counterparts

Figure 6 demonstrates that CATs scale similar to their dense transformer equivalents. We evaluate against three dense transformer scales $\{31M, 92M, 260M\}$, with their CAT equivalents containing parameters $\{95M, 326M, 1B\}$. All models were trained for 15B tokens, under the setup in Section 3.

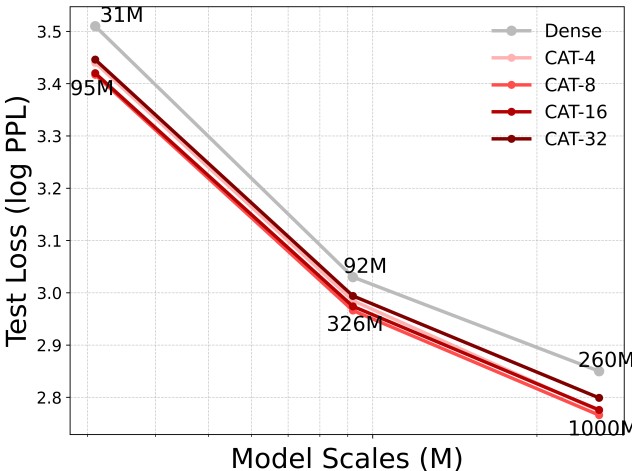

*Figure 6.* CATs scale like their dense transformer counterparts while being up to $3\times$ faster and $9\times$ more memory-efficient. All CAT curves come from a single model, evaluated at different chunk sizes. **Note that while CAT occupies more parameters, it is still both compute and memory efficient compared to the densee transformer at every scale.**

## B.14. Across chunk analysis

We provide how the validation loss changes within a chunk in Figure 7. We provide averaged results across all chunks. We provide different curves for each chunk size.

Interestingly, across all chunk sizes, the loss is highest when decoding the first token from the compressed representations only. After that token is decoded, the loss decreases steadily as CAT keeps decoding tokens from both compressed representation and raw tokens that appear before inside the chunk.

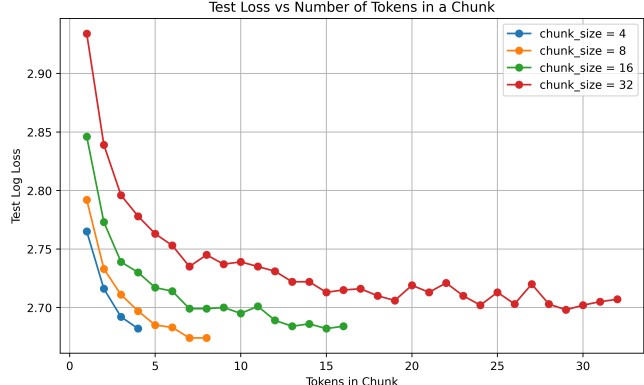

*Figure 7.* Chunk Analysis

## B.15. Finetuning CATs on S-NIAH-U

S-NIAH-U is a task where model needs to recall 32 token long UUID strings from the long context. This section reports performance of CATs after task specific finetuning on samples from S-NIAH-U. We only apply the loss on tokens that appear in the answer span. Table 14 reports these results. This is accompanied by loss curves for different CATs depending on chunk size in Figure 8 on this task.

We observe two things: (i) after finetuning, performance goes up significantly for all chunk sizes. This signifies as chunk size increased, compressor in CATs, before finetuning, was not surfacing the *right* information in the chunk representation. (ii) the loss curves during finetuning indicate the same as well, however it still does not go completely to zero, especially for CAT-32. This indicates that there are limits to what information a fixed sized chunk representation can practically learn to surface, justifying its sub-par accuracy on the task.

This problem of not surfacing the *right* information in the chunk representation could be alleviated by more and longer pre-training, or choosing smaller chunk sizes for tasks that require accurate recall.

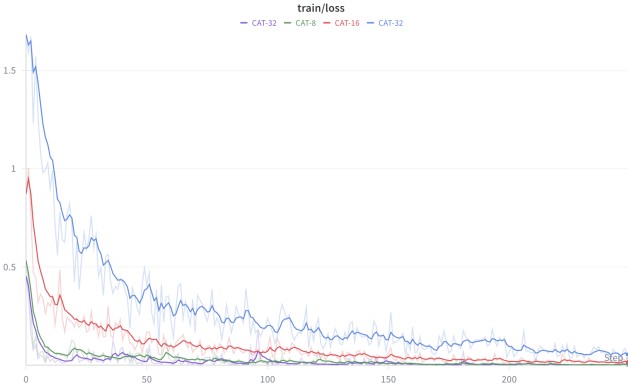

*Figure 8.* Loss curves when finetuning different CATs on samples from S-NIAH-U task.

| Model | Before | After |
|-------|--------|-------|
| CAT-4 | 46.5 | 97.1 |
| CAT-8 | 47.3 | 97.0 |
| CAT-16 | 3.8 | 94.2 |
| CAT-32 | 0.0 | 64.3 |

*Table 14.* Performance on 4K sequence length before and after finetuning for different CAT variants.

# C. Implementation details and PyTorch style pseudo-code

In this section, we discuss some implementation details regarding CATs. We repeat some text presented in the main paper to be self-contained below.

## C.1. Training

**Training:** While CATs are simple and build on dense transformer abstractions, their naive PyTorch training implementation is very inefficient.

Note that compression of chunks of tokens is efficient since it can be done in parallel, specifically using $\texttt{torch.vmap}(f_\theta(\mathbf{c}_i))$ for all chunks $\mathbf{c}_i$. This costs a total of $O(\frac{N}{C} \cdot C^2) = O(NC)$ in self-attention compute, which is much better than $O(N^2)$.

But, computing logits for tokens in chunk $\mathbf{c}_i$, that is computing $g_\theta(\mathbf{c}_i \mid f_\theta(\mathbf{c}_1) \cdots f_\theta(\mathbf{c}_{i-1}))$ can be non-trivial since for chunk $\mathbf{c}_i$, we have $i-1$ past chunk representations $\{f_\theta(\mathbf{c}_1), f_\theta(\mathbf{c}_2) \ldots f_\theta(\mathbf{c}_{i-1})\}$. In other words, there are different number of past chunk representations for every chunk, making shapes variable and as a result, harder to parallelize computation of logits. One could employ a python loop and compute logits for every chunk sequentially, but that would be slow and won't scale. In fact, even if one manages to compute logits for every chunk in parallel, the total self-attention operations in the decoder would be $O(\sum_{i=1}^{\frac{N}{C}}(i+C)^2) = O((\frac{N}{C})^3)$, that is cubic in sequence length. Padding to make shapes constant would make things worse. Thus, naive techniques will not scale.

*With such difficulties in making the training scalable, it may not be surprising that despite the simplicity of CATs, it was not attempted in the community.* Note that unlike CATs, similar architectures (Ho et al., 2024; Yu et al., 2023) do not have this problem: computing logits can be naively parallelized due to fixed shapes and self-attention operations scale quadratically due to a single compressed representation for the past.

In CATs, observe that in computing logits chunks $\mathbf{c}_i, \mathbf{c}_{i+1} \ldots \mathbf{c}_{\frac{N}{C}}$, one calculates the same key-values for chunk representations $f_\theta(\mathbf{c}_j)$ in the decoder, where $j < i$. This points to repeated and identical computations. To exploit this observation, we take advantage of a custom attention mask in decoder to calculate logits for all chunks in parallel, and reuse computations done for a past chunk representation to be used for a computations for logits for a future chunk. To be concrete, once we calculate all chunk representations $f_\theta(\mathbf{c}_i)$ in parallel using $\texttt{torch.vmap}$, we insert $f_\theta(\mathbf{c}_i)$s at particular positions in the original sequence: after every chunk $\mathbf{c}_i$, we attach its chunk representation. That is, sequence would look like: $\{\mathbf{c}_1, f_\theta(\mathbf{c}_1), \mathbf{c}_2, f_\theta(\mathbf{c}_2), \ldots \mathbf{c}_i, f_\theta(\mathbf{c}_i) \ldots \}$. Now, we pass this sequence into the decoder during training, with a custom attention mask (see Figure 9) that allows a token in chunk $\mathbf{c}_i$ to attend to previous tokens within that chunk only as well as only to previous chunk representations, which would be $f_\theta(\mathbf{c}_{i-1}), f_\theta(\mathbf{c}_{i-2}) \ldots f_\theta(\mathbf{c}_1)$ only. Any token in chunk $\mathbf{c}_i$ does not attend to raw tokens outside this chunk. This implementation allows re-use of key-values for chunk representations $f_\theta(\mathbf{c}_i)$ for calculation of logits of future chunks, in parallel, making the training of CATs efficient and scalable. We utilize the FlexAttention API (Dong et al., 2024) to automatically create a custom kernel for the custom mask (Figure 9). Note that this way of computing logits is quadratic in sequence length but with a constant times better: concretely it is $O(\frac{N}{C} \cdot N + \frac{N}{C} \cdot C^2) = O(\frac{N^2}{C})$, **which is** $C\times$ **better than** $O(N^2)$ (yellow dots in figure 9 provides a visual proof for this cost; number of yellow dots are significantly lower than $\frac{N^2}{2}$). Mathematically the cost of attention in CATs decoder is: $\sum_{i=1}^{N}[\frac{i}{C}] + (i \bmod C) + 1 = O(\frac{N^2}{C})$, where $[.]$ is the floor function, and $\bmod$ is modulo operator.

For a discussion in training throughput, refer to a discussion in Appendix C.5.

```
def forward(input_ids, targets):

    input_ids = einops.rearrange("b (k c) -> b k c", k=num_chunks, c=chunk_size)

    # calculate f(x)
    # shape of fx: (b, k, D_d)
    fx = torch.vmap(f)(input_ids)

    output_logits = list()
    for i in range(num_chunks): # note that this loop is done in parallel with the custom
        attention mask presented in the appendix
        # use the previous i+1 fx to predict the current chunk
```

```
13        # shape of cur_chunk_logits: (b, 1, l, V)
14        cur_chunk_logits = phi(input_ids[:, i, :], fx[:, :i+1, :])
15        output_logits.append(cur_chunk_logits)
16    output_logits = torch.cat(output_logits, dim=1) # shape: (b, k, c, V)
17    output_logits = einops.rearrange(output_logits, "b k c v -> b (k c) v") # arrange all
          chunks logits together (or flatten)
18    return torch.nn.functional.cross_entropy(output_logits, targets) # return the loss
```

*Listing 1.* Pseudocode for training step

## C.2. CAT's training attention mask

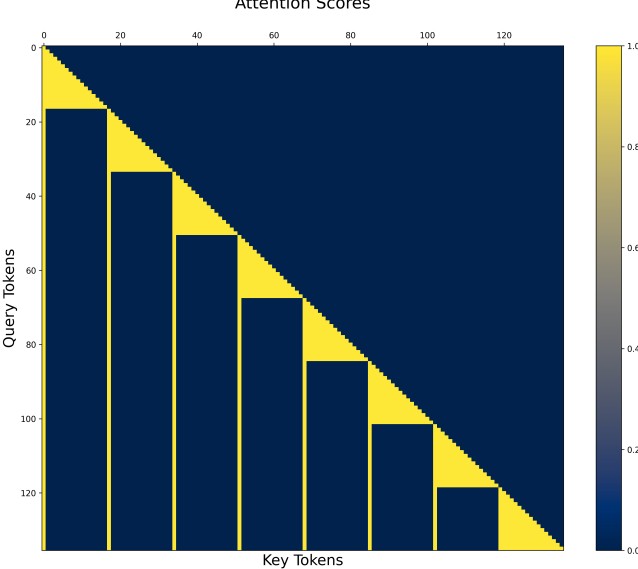

*Figure 9.* Sequence length is 128, and the chunk size that we use in this particular attention mask is $C = 16$.

Note that attention mask in figure 9 looks very similar to the attention mask as defined in (Child et al., 2019), however, in CAT's case: (a) it is not heuristic choice, and (b), tokens in a particular chunk attend to the past $f_\theta(\mathbf{c}_i)$ representations obtained by the compressor, rather than the past token embeddings at that position as done in (Child et al., 2019).

## C.3. Generation

The decoder during generation attends to atmost $\frac{N}{C} + C$ tokens. Due to compression, CATs can throwaway past chunks of tokens, and only keep their compressed chunk representations in memory. This straightaway results in a big reduction of memory; the KV cache is slashed by a factor of $C$. For even a moderate chunk size of 4, this results in big reductions in memory during generation (Figure 2) compared to a dense transformer. This slash in memory is accompanied by reduced memory accesses a decoder makes in CATs, which is the major bottleneck during generation. Costs for self-attention in CATs decoder scale as $O(\frac{N^2}{C})$, which is again, $C\times$ better than $O(N^2)$ for a dense transformer.

Implementing generation is simpler than training and very similar to how it occurs for a dense transformer. In fact, a pure PyTorch implementation for CATs is on-par with efficient architectures that utilize custom kernels. We inspire our implementation from: https://github.com/meta-pytorch/gpt-fast. Given $i$ chunks of tokens: firstly, torch.vmap over chunks independently to calculate $f_\theta(\mathbf{c}_i)$ in parallel. Then prefill the decoder's KV cache in parallel with the obtained $f_\theta(\mathbf{c}_i)$s. Now generate the next chunk $\mathbf{c}_{i+1}$ autoregressively one token at a time. Note that this uses a simple causal mask since the previous positions are already prefilled with $f_\theta(\mathbf{c}_i)$s, which is required to decode chunk $\mathbf{c}_{i+1}$. Once all the tokens of the chunk $\mathbf{c}_{i+1}$ are generated, calculate $f_\theta(\mathbf{c}_{i+1})$ and prefill the decoder's KV cache just after the position where $f_\theta(\mathbf{c}_i)$ was cached. Now the KV cache is ready for generation of the next chunk $\mathbf{c}_{i+2}$ and this process will continue.

This simple implementation enables CATs to be $1.4 - 3.2\times$ **faster** than the dense transformer while showcasing **upto** $2.2 - 9.5\times$ **lower total memory usage** as one increases chunk sizes.

```python
# https://github.com/pytorch-labs/gpt-fast/blob/7dd5661e2adf2edd6a1042a2732dcd3a94064ad8/
    generate.py#L154
def generate_chunk_by_chunk(
    input_ids
):
    # assume input_ids.shape == (batch_size, 1, chunk_size)

    # declare/reset static KV cache, shape: [batch_size, num_chunks + chunk_size, 2, D_d]

    input_pos = 0

    # compress the first chunk (batch_size, 1, chunk_size) -> (batch_size, 1, D_d)
    # get fx for the very first chunk
    fx = f(input_ids) # shape of fx: (batch_size, 1, D_d)
    next_token = prefill(fx, input_pos) # prefill at idx 0 with fx in phi

    new_chunks = list()

    for i in range(num_chunks - 1):

        # generate entire chunk using fx that was prefilled earlier in phi
        next_chunk = generate_chunk(next_token)
        new_chunks.append(next_chunk.clone())

        # get new fx
        # compress the new obtained chunk
        fx = f(next_chunk) # (batch_size, 1, chunk_size) -> (batch_size, 1, D_d)

        # prefill again at input_pos
        input_pos += 1
        next_token = prefill(fx, input_pos) # prefill fx at idx `input_pos` in phi

    new_chunks = torch.cat(new_chunks)
    return new_chunks
```

*Listing 2.* Pseudocode for generation

### C.4. Adaptive CATs training details

To enable training of adaptive CATs, we made some choices that we now describe. In every training iteration, we sample a chunk size uniformly at random and perform loss computation. Further, due to variable size of a chunk in every training iteration, one cannot keep a single projection matrix that projects processed token embeddings in the compressor to a single chunk representation (since shapes for projection matrix would be different for different chunk size). One could tackle this by keeping an independent projection matrix for every chunk size, but we found this didn't work well empirically, possibly due to reduced updates for every chunk size's projection weights (only one chunk size's projection weights are updated per iteration; this is not the case with compressor or the decoder, they are updated every iteration). Instead, we took inspiration from (Beyer et al., 2023) where the authors declared a single projection matrix for all chunk sizes, and then linearly interpolated the matrix to the desired shape depending on the current chunk size. This means the linear interpolation is also under `torch.autograd` and is optimized so that the final linearly interpolated projection matrix gives a *good* chunk representation for every chunk size.

### C.5. CAT's training throughput analysis

We make use of FlexAttention API to obtain a custom self-attention kernel specifically for the masking scheme section 9. This fused kernel gives a significant boost in training throughput in self-attention costs compared to using a naive PyTorch masked implementation.

MLPs in a transformer drive the majority of the FLOPs budget during training at smaller sequence lengths (Austin et al., 2025). At a sequence length of 4096, CATs take $\leq 2.35\times$ to train compared to a dense transformer (measured on batch size of 8 with compressor depth of 3, decoder depth of 6, hidden size for compressor $D = 1024$ and hidden size for decoder $D_g = 2D = 2048$ for CAT, compared against dense transformer having depth of 6 and $D = 1024$, on a A100 80 GB PCIe.) At 32K sequence length, this gap reduces significantly and CAT only costs $1.5\times$ more.

Despite this, CAT amortizes this overhead in two ways: (i) through cheaper inference, which dominates lifetime cost, and (ii) by replacing multiple models with one — training independent models to cover the same range of inference budgets would require separate pretraining runs for each operating point, costing more than a single CAT.

### C.6. Time taken by each CAT component during generation

Here we measure time taken by each CAT component during generation, specifically time taken by: decoder attention, decoder FFNs, and time taken by parallel compression. Section C.6 provides these results. We use the same setup in benchmarking as described in Section 3. We use a chunk size of 8 for this ablation.

| Component | Time (ms) | Percentage (%) |
|---|---|---|
| Attention in Decoder | 30,817 | 70.1 |
| FFN in Decoder | 11,555 | 26.3 |
| Compression in Compressor | 1,551 | 3.5 |
| Total | 43,932 | 100.0 |

# D. Some ablations on the CAT architecture

## D.1. Ablation on hidden size of compressor

With this ablation, we show that increasing hidden size of the compressor does not help in improving perplexity. We fix $D_g = 1536$ for these experiments. For this ablation, we use a smaller WikiText-103 dataset. Both compressor and decoder use the same depth $L = 6$.

| Chunk Size $C$ | Size of $D_f$ | Perplexity |
|---|---|---|
| 16 | 768 | 17.6 |
| | 1536 | 17.6 |

*Table 15.* Comparison of choices of hidden size of compressor on WikiText-103 perplexity.

There is no effect of increasing the hidden size of the compressor. The performance before and after remains the same.

## D.2. Ablation on hidden size of decoder

We ablate on different choices of $D_g$ along with different chunk sizes in CAT . In this setup, we fix $D_f$ in the compressor, and only vary $D_g$ or $C$ (chunk size). We use WikiText-103 for these experiments. In this setup, $D = 768$. Both compressor and decoder use the same depth of $L = 6$.

| Chunk Size $C$ | Size of $D_g$ | Perplexity |
|---|---|---|
| 4 | $D$ | 19.8 |
| | $2D$ | 17.4 |
| 8 | $D$ | 20.4 |
| | $2D$ | 17.7 |
| 16 | $D$ | 20.2 |
| | $2D$ | 17.6 |

*Table 16.* Comparison on choices of chunk sizes and sizes of $D_g$ on WikiText-103 perplexity.

We observe that we obtain the best perplexities when we $D_g = 2D$ for the particular chunk size we are using. Using this observation, we used this as our *default* configuration for the FineWeb-Edu experiments.

| Model | $D_f$ | $D_g$ | Perplexity | Avg. recall |
|---|---|---|---|---|
| Dense | $--$ | $D$ | 21.2 | 23.8 |
| CAT | $D$ | $D$ | 23.8 | 13.7 |
| CAT | $D$ | $2D$ | **20.7** | **19.8** |

*Table 17.* Impact on perplexity and average recall performance of CAT when varying $D_g$. For dense, $D_g$ implies hidden size for itself. Here, $D = 1024$. $D_g = 2D$ gives better perplexity and average recall. We train CAT only at chunk size $C = 8$ for these experiments. All models were trained for 5B tokens with 1K sequence length. Rest of the setup follows Sec. 3.

## D.3. Ablation on depth of the compressor

We ablate on the depth of the compressor. For a fixed chunk-size, $D_f = 768$ (compressor embedding size), $D_g = 1536$ (decoder hidden size), and a fixed depth of the decoder, we vary the compressor depth.

| Chunk Size $C$ | Depth of Compressor | Perplexity |
|---|---|---|
| 8 | 6 | 17.4 |
| | 3 | 17.4 |
| 16 | 6 | 17.8 |
| | 3 | 17.7 |

*Table 18.* Comparison on choices of depth of the compressor across different chunk sizes $C$ on WikiText-103.

We have an interesting observation that one can reduce the depth of the compressor without sacrificing on the downstream perplexity. This could mean one can compress small chunks of tokens without a requiring high capacity. In our generation benchmarks, we observed that compressor depth play less of a role in latency as compared to the decoder depth (since we compress tokens in parallel using one transformer call). That being said, compressor depth does play a significant role in training costs (due to the MLP training costs in the compressor). Therefore, reducing compressor depth goes into overall advantage for the CAT architecture.

However, what is the limit, and can one go to even a 1 layer of compressor is an interesting question to ask. There might be some lower bound on the compressor depth to start compressing chunks of tokens, but we leave this to future work.

# E. More experiment details

Here we provide more details about the experiments done in the main text.

## E.1. Baselines

In this section, we provide details about the models used in our experiments.

| Model | Total (M) | Embedding (M) | Non-Embedding (M) |
|---|---|---|---|
| Dense | 260 | 50 | 210 |
| Dense-D/2 | 92 | 25 | 70 |
| Mamba2 | 260 | 50 | 210 |
| GDN | 310 | 50 | 260 |
| GDN-2x | 310 | 50 | 260 |
| GDN-Hybrid 1:1 | 280 | 50 | 230 |
| GDN-Hybrid 1:1 D/2 | 111 | 25 | 86 |
| GDN-Hybrid 1:4 | 280 | 50 | 230 |
| GDN-Hybrid Global 1:7 | 300 | 50 | 250 |
| GDN-Hybrid Global 1:7 2D | 820 | 100 | 720 |
| Sparse-4 | 820 | 100 | 720 |
| Sparse-8 | 820 | 100 | 720 |
| CAT-4/8/16/32 | 150 + 820 | 50 + 100 | 100 + 720 |

*Table 19.* Model parameter sizes in millions, separated into embedding and non-embedding parameters. Parameters for CATs consists of parameters in compressor + parameters in decoder.

By default, all models below are configured with $L = 12$ layers and $D = 1024$ hidden dimension; any deviations are explicitly stated.

1. Dense transformer (or Transformer++) (Vaswani et al., 2017; Touvron et al., 2023): We use rotary position embeddings along with the FlashAttention kernel to perform self-attention. The MLP is a SwiGLU MLP (Touvron et al., 2023). Dense-D/2 uses $2\times$ lower model dimension of $D = 512$.

2. Sparse transformer (Child et al., 2019): Follows the Dense transformer configuration, except the attention mask used. Moreover, we used $D = 2 \cdot 1024 = 2048$ for this baseline for a fair comparison with CATs. We used FlexAttention API to create optimized Flash Attention like kernel for this. We use a stride length of 4 (Sparse-4) and 8 (Sparse-8) that tries to compete with CAT-4 and CAT-8 respectively.

3. MAMBA2 (Dao & Gu, 2024): The model uses 2 Mamba mixer per layer. All layers use the MAMBA2 block without any mixing any attention. The `expand` is set to 2, $d_{state} = 128$, and convolution $k = 4$. Activations used are SiLU. We use the official codebase for MAMBA2 generation throughput and memory benchmarking: https://github.com/state-spaces/mamba and code from: https://github.com/fla-org/flash-linear-attention for training.

4. GatedDeltaNet (GDN) (Yang et al., 2025b): We use the implementation provided at https://github.com/fla-org/flash-linear-attention for training. We use `head_dim` as 128 and `num_heads` as 8 (same as MAMBA2 above). GDN-2X stands for recurrent state size increased by $2\times$.

5. GatedDeltaNet Hybrids (GDN-H) (Yang et al., 2025b) We insatiate multiple GatedDeltaNet Hybrids models in our comparison. GDN-H 1:1 uses use sliding window layers at every other layer with a sliding window size of 2048. GDN-H 1:4 uses the same sliding window layers, but in the ratio of $1 : 4$ with linear attention. GDN-H 1:1 D/2 uses the same sliding window layers at every other layer, but the model dimension is scaled down by $2\times$ to $D = 512$.

   Finally, GDN-H 1:7 G uses linear-dense attention ratio as 1:7 but with a global attention. GDN-H 1:7 2D G uses linear-dense attention ratio as 1:7 with global attention and uses $2\times$ the hidden size.

### E.2. Datasets

Following common practices done in (Gu & Dao, 2023; Dao & Gu, 2024; Arora et al., 2024a; Yang et al., 2025b), we evaluate all models on multiple common sense reasoning benchmarks: PIQA (Bisk et al., 2020), HellaSwag (Zellers et al., 2019), ARC-challenge (Clark et al., 2018), WinoGrande (Sakaguchi et al., 2021) and measure perplexity on WikiText-103 (Merity et al., 2016)and LAMBADA (Paperno et al., 2016). In Table 7, HS denotes HellaSwag, PQ denotes PIQA, AE denotes ARC-Easy, AC denotes ARC-Challenge, WG denotes Winogrande, OQA denotes OpenBookQA, LMB denotes LAMBADA, Wiki denotes WikiText, and FW denotes FineWeb-Edu.

We evaluate on tasks from LongBench (Bai et al., 2023) where each abbrevation in table 5 stands for: QAS: `qasper`, MQA: `multifieldqa_en`, HQA: `hotpotqa`, 2WMQ: `2wikimqa`, TQA: `triviaqa`, TREC: `trec` split of LongBench.

To measure real-world recall accuracy, we use datasets used in (Arora et al., 2024a;b). Namely these consists of SWDE (Lockard et al., 2019) for structured HTML relation extraction and several question answering datasets including SQuAD (Rajpurkar et al., 2018), TriviaQA (Joshi et al., 2017), DROP (Dua et al., 2019) and FDA (Arora et al., 2023b). Since our pretrained models are small, we use the Cloze Completion Formatting prompts provided by (Arora et al., 2024b).

We evaluate on tasks from the needle-in-haystack benchmark RULER (Hsieh et al., 2024).

Additionally, we evaluate on datasets from the LongBench benchmark (Bai et al., 2023) to evaluate long-context understanding.

Finally, to evaluate baselines on state-tracking tasks, we used the BabiLong benchmark (Kuratov et al., 2024). Due to relatively small scale of our setup, we were only able to evaluate on `qa1` subset, since for other complex subsets, all baselines failed.

### E.3. Generation

Both dense transformer and CAT use FlexAttention API causal dot product kernels. We use the script provided in (Dao & Gu, 2024) to benchmark[6] Mamba2, GatedDeltaNet and GatedDeltaNet-Hybrid. All benchmarks used a prefill of 8 tokens. All benchmarks were run using a single NVIDIA A100 80GB PCIe, and use CUDA cache graphs for the next-token prediction.

### E.4. Main figure details

Figures 1b and 5 reports memory usage at 2K sequence length since both SWDE and FDA datasets have queries with context length upto 2K. The latencies in Figure 5 are reported at maximum sequence length of 4K.

---

[6]github.com/state-spaces/mamba

## F. Extended Related Work

**Reducing self-attention costs:**   Reducing the cost of self-attention enables scaling transformers to large contexts and has been the focus of much work Child et al. (2019); Parmar et al. (2018); Beltagy et al. (2020); Jiang et al. (2023). Common techniques include *heuristically* defined sparse attention maps (Child et al., 2019; Zaheer et al., 2020) or a sliding window (Jiang et al., 2023) in order to reduce the tokens being attended to. The compute required (and in some cases, memory) for attention go down, however, compromising with the expressivity of the model. In turn, to achieve performance similar to that of full-attention, efficient models either require big window sizes (making their memory costs large again) (Arora et al., 2024a) or more layers (in case of sparse or sliding window attention, see App. B.9 and Tab. 1).

(Shazeer, 2019) proposes use of single or reduced key and value heads in the self-attention block, more commonly known as Grouped Query Attention (only one key/value head) or Multi Query Attention (reduced key/value heads). This results in reduction of memory with seemingly no loss in downstream performance, making this a popular choice in latest model releases (Yang et al., 2025a). That being said, one could use the same technique inside CAT's decoder (and compressor) self-attention block, making it complimentary.

Concurrent works like (Yuan et al., 2025) reduce attention compute by attending to compressed past tokens as well as to specific blocks of uncompressed tokens in the past. This is similar in spirit to our work, however, in the case of (Yuan et al., 2025), there are no memory savings during inference.

Some works (Rae et al., 2020; Chevalier et al., 2023) explored recurrent formulations of a transformer to enable processing of longer sequences on limited compute by compressing past context. However, training sequence models in a recurrent fashion has its own challenges, back-propagation through time (BPTT) being the most important one. More recently (Geiping et al., 2025) had to use very careful weight initialization, truncated gradients, small learning rates and careful placement and tuning of norms to train a large-scale recurrent architecture in a stable manner and prevent optimization collapse. Nevertheless, these techniques are complementary to CAT.

Alternatively, one can optimize the computation of full-attention to directly reduce wall-clock time and memory by leveraging hardware advancements. For example, Dao et al. (2022) compute attention in blockwise manner and exploit the nature of online softmax (Milakov & Gimelshein, 2018) which removes the need to instantiate the entire $QK^T$ matrix and reduce calls to slow-read part of the GPU memory. As we utilize the attention mechanism as is, any reductions in cost due to hardware optimization that apply to the attention mechanism also proportionally reduce the cost of CAT models.

Finally, plethora of works have tackled reducing compute and memory requirements of a transformer in a *post-hoc* manner i.e. after it has been trained using full-attention (also called *training-free* sparse attention (Nawrot et al., 2025)). Common techniques include prefill-time sparsification (vertical/slash/block; adaptive) and decode-time KV-cache selection/eviction (e.g. (Li et al., 2024b; Tang et al., 2024)). However, because models are trained dense but run sparse, train–test mismatch can hurt downstream performance. Still, these works are orthogonal to CAT and can be layered on CAT's decoder, making them complementary.

**Linear attention and state-space models:**   A different line of work reduces the generation cost of transformers by limiting the recurrent state, which is the vector required to decode each token. Self-attention keeps track of the entire context (or the KV cache) meaning that the recurrent state increases in size with each decoded token. Works like (Arora et al., 2024a; Katharopoulos et al., 2020) linearize attention to make a fixed-size recurrent state that can be updated via simple averaging; the technique is to approximate self-attention with linear operations of query, key, and value vectors transformed through a feature map. The choice of the feature map falls to the user and approximating attention well requires the feature map to be large in size, which can counteract the gains in computational costs achieved by the linearization.

Alternatively, one can replace attention with linear or pseudo-linear sequence mixers such as state-space models (SSMs) (Gu et al., 2021; Sun et al., 2023), gated convolutions (Fu et al., 2022; Poli et al., 2023) and input-dependent recurrent (Peng et al., 2023; Gu & Dao, 2023) and more recently (Yang et al., 2025b).

Typical implementations of linear attention and state-space models do achieve impressive reductions in generation costs and memory, but restrict the expressivity to the extent that these models do not solve in-context recall tasks without large recurrent state sizes (Arora et al., 2024a; 2023a), or without composing with other sequence mixers, such as local sliding window attention (Arora et al., 2024a; Yang et al., 2025b). Choosing such a composition again falls back to the user, complicating the design process. Additionally, this process trades-off computation costs for performance because the attention layers that improve recall performance also come with larger time and memory costs.

Unlike the works discussed above, CATs require no complicated changes to the attention mechanism itself. CATs rely on the fact that natural language is redundant and can be compressed (Zipf, 2016; Shannon, 1951). Instead of relying manual approximations of history or utilizing any heuristic choice for feature maps, we let the model and optimization decide what the history should be using learned compression. Moreover, its unclear how much memory and compute a downstream task requires, making the adaptive property of CATs much desirable, which no other baselines provide.

**Hierarchical transformers:** Many previous works (Pappagari et al., 2019; Han et al., 2021; Dai et al., 2020) have explored employing hierarchy in transformers for creating representations representations for documents/images, where a *local* encoder transformer processed parts of the document/image independently. Later works (Nawrot et al., 2021; 2022; Slagle, 2024) explored downsample-then-upsample approach (*hour-glass* like structure), where the sequence is downsampled into *coarse* tokens followed by upsampling into *fine-grained* tokens before being decoded. Due to the *hour-glass* structure, there are compute savings during training, but during generation, the architecture must maintain a cache for all the past tokens, leading to significant memory accesses. Concurrently, (Hwang et al., 2025) explored a dynamic and end-to-end learned strategy for chunking in *hour-glass* like architectures.

Different from above, works like (Ho et al., 2024; Yu et al., 2023) break up the modeling of a sequence into chunks/patches, where each chunk is modeled independently of each other given the previous "global" chunk embedding. An embedder first compresses each chunk independently, then these "local" chunk embeddings are passed to a "global" model where each "local" chunk embedding attends to past "local" chunk embeddings, forming a "global" chunk embedding. Each "global" chunk embedding is then passed to a decoder that is responsible for generating the next chunk.

On first glance, CATs might appear similar to above works, specifically (Ho et al., 2024; Yu et al., 2023), however the subtle but salient difference is: one directly feeds **all the previous "local" chunk/patch representations** directly to the decoder in CAT, whereas in works like (Ho et al., 2024), one feeds in just the previous "global" chunk representation outputted by a "global" model to the decoder. This architectural choice of passing *all* the compressed local chunks from the past directly to the decoder allows CATs to solve long-range recall tasks with ease while maintaining efficiency, whereas (Ho et al., 2024) is plagued by *learnability* problems (even in toy recall tasks) due to constant size compression of history. Additionally, CATs don't utilize three different components (embedder, global decoder, local decoder) – it only uses a compressor and a decoder, reducing the design space and (significant) parameter requirements further.

Additionally, (Yen, 2024) extend the cache by using a modified encoder-decoder architecture, where decoder attends directly to final activations of a smaller fixed encoder, without any compression.

Finally, (Barrault et al., 2024) suggest learning "concepts" instead of tokens by modeling the latent representation of language produced by pushing the token sequence through a large sentence embedder. The focus of this work is to decouple the modeling of the low-level details in each language, like tense and grammar, from the larger concept space that is shared across languages. In contrast, the goal with CAT is to reduce the cost of modeling sequences and can be used as a plug-and-play replacement to the latent concept model. Moreover, the encoder in (Barrault et al., 2024) is an auto-encoder, that might keep irrelevant information in the chunk representation. Compressor in CATs only keeps information that is predictive of the future chunks.

**Adaptive architectures:** (Kusupati et al., 2022; Devvrit et al., 2023) learns representations during training time that can work at different granularity during test-time, yielding adaptivity to the learned architecture. However, coarser granularity of *Matryoshka* representations result in loss of language modeling performance (in terms of perplexity) (Devvrit et al., 2023). That being said, one could apply similar approaches to CATs making them complimentary. CATs use the same high-level approach described in (Beyer et al., 2023): learn a single model that can work for various patch sizes at once depending on the downstream use-case at test-time. However, (Beyer et al., 2023) worked with image classification tasks; CATs deal with language modeling and generation.

