# OpenReview forum: "Compression is all you need for Controllably Efficient Language Models"
_ICML.cc/2026/Conference — Submitted to ICML 2026_

### Official Review · Reviewer_F7bn · 2026-03-11

**Soundness:** 2
**Presentation:** 3
**Significance:** 3
**Originality:** 2
**Overall Recommendation:** 3
**Confidence:** 4

**Summary:**

The manuscript introduces the Compress & Attend Transformer (CAT), a meta-sequence mixer designed to mitigate the high inference memory costs of dense attention while preserving strong in-context recall. CAT partitions sequences into uniform chunks, utilizes a "compressor" to condense each chunk into a single representation vector, and employs a "decoder" that attends to these compressed representations alongside local uncompressed tokens. Experimental results indicate that CAT matches or outperforms dense transformers and efficient linear/hybrid baselines on long-context recall and understanding benchmarks while significantly reducing memory footprint and generation latency.

**Compliance With Llm Reviewing Policy:**

Affirmed.

**Final Justification:**

Some questions are clarified, but considering the limited novelty and experimental deficiencies, I will keep my scores unchanged.

**Key Questions For Authors:**

N/A

**Limitations:**

yes

**Strengths And Weaknesses:**

Strengths:
- The proposed CAT architecture elegantly decouples the sequence length bottleneck by introducing a chunk-level compression step followed by causal decoding, offering a theoretically sound memory reduction mechanism without restricting global context access.
- The parallel training implementation is technically impressive; by restructuring the input sequence to interleave raw tokens and compressed representations and utilizing a custom FlexAttention mask, the authors successfully bypass the slow sequential optimization typical of recursive compression models.

Weakness:
- Unfair Comparisons due to Capacity Mismatch. The main text evaluations are fundamentally misaligned because the proposed CAT model utilizes significantly more total parameters than the dense baselines (as revealed in Appendix E.1, Table 19). It is well-established that MoE models[1] inherently outperform dense models when active parameters are held constant. Despite supplementary experiments later in the text, the core results fail to isolate CAT's algorithmic contribution from its structural capacity advantage.
- The assertions regarding CAT's effectiveness in the paper are overstated. Specifically, the empirical results presented in Appendix Tables 6 and 7 do not robustly support the authors' strong claims.
- Limited Novelty and High Training Cost. The technical contribution is limited given the existence of similar compression methods (e.g., [2], [3]). These prior works achieve comparable goals efficiently by fine-tuning pre-trained models. In contrast, CAT's requirement to train from scratch introduces unnecessary complexity, introducing severe computational costs that limit its broader practical adoption.

Reference:
- [1]. Deepseekmoe: Towards ultimate expert specialization in mixture-of-experts language models. 2024.01
- [2]. LightThinker: Thinking Step-by-Step Compression. 2025.09
- [3].Token Assorted: Mixing Latent and Text Tokens for Improved Language Model Reasoning 2025.02

---

> ### Author Rebuttal · Authors · 2026-03-31
>
> > W1: capacity mismatch
>
> We define capacity as the number of distributions that can be represented by the model.
>
> Firstly, lots of parameters does not mean high model capacity: as an extreme example, a model consisting of only an embedding and an un-embedding layer with an arbitrarily large embedding dimension has lots of parameters, however it can only represent token bigrams (limited distributions). Thus, the model architecture *matters* for model capacity.
>
> Now, within a fixed model architecture, more parameters typically mean more capacity. However, across architectures, this need not hold. In fact, without downstream evaluations, its not immediately clear if MoEs or CAT have higher or lower capacity than dense transformer since they are fundamentally different model architectures.
>
> We argue what ultimately matters is how the model architecture and number of parameters interact to yield high model capacity at the same inference cost -- because this would imply better results as the reviewer suggests, without using more costs. Capacity can be changed for a fixed inference cost by changing model architecture and parameters.
>
> In fact, matching active parameters controls for inference cost, not capacity, and indeed, the whole point of MoEs is that they provide higher capacity (better results) at the same inference cost, which is a desirable property, not a confound.
>
> The same logic applies to CAT: compression of sequence enables more parameters (wider decoder) without proportionally increasing inference costs. This is a feature and contribution of CAT architecture, not a flaw in the comparison.
>
> If the reviewer has an alternative definition of capacity, we’d happily engage.
>
> > W2: isolate CAT's algorithmic contribution
>
> CAT has three distinct contributions:
> - **More parameters at lower inference costs:** We discussed this above.
> - **Flexible memory that grows gracefully with sequence length:** due to compression, the memory (KV cache) for CAT grows with sequence length, but by a factor less (O(N/C)) depending on the chunk size -- **providing a middle ground between fixed memory (Mamba2, GDN) and full memory (Dense).** We argue this flexible memory helps with long-context recall and modeling performance of CAT. See Table 4 for which architectures have both flexible and efficient memory – CAT is unique in satisfying both.
> To isolate this: please refer to: *(i) App. B.5:* other sequence mixers given identical memory as CAT fail on simple recall task, exposing fixed-memory limitations; *(ii) App. B.6:* Block Transformer with same or more parameters and memory than CAT still fails, indicating more parameters alone do not explain CAT's performance; *(iii) Fig. 1:* CAT-4 outperforms Sparse-4/8 and GDN-H 1:7 G 2D on recall at similar parameters and matched inference cost.
> - **Test-time controllable inference costs.** A single model exposes multiple inference costs -- no other sequence mixer provides this.
>
> > W2
>
> Tables 7 and 8 (appendix) report averaged perplexity and short-context benchmarks (~30 tokens). While necessary, these are insufficient for evaluating long-context ability (averaged perplexity does not correlate with long-context recall).
>
> Nevertheless, CAT performs competitively in both, outperforming all except one (stated explicitly in paper).
>
> In long-context **real-world** recall (Fig. 1), CAT matches or surpasses different models at a given inference cost using a using a **single** model only, whose inference cost can be changed, supporting our claims in the abstract and the introduction.
>
> > W3: Limited Novelty
>
> There seems to be a misunderstanding here. References provide by the reviewer [2, 3] do not provide controllable efficiency at test time in a single model. One would have to train/finetune a separate model for each efficiency level.
> In light of this, if the reviewer can clarify what they mean by comparable goals, we'd be happy to discuss more. We will cite and discuss these in our related work.
>
> Further, CAT fundamentally is a novel *meta*-sequence mixer that uses existing off-the-shelf sequence mixers as its components (refer to Table 4 in the appendix) – this means it can be used to model any sequence e.g. text, audio, video, DNA etc. -- unlike the works referenced that work potentially only on text data.
>
> > W3: High training cost
>
> Pre-training from scratch is standard for model architecture or sequence mixer contributions. Since CAT is a meta-sequence mixer, we pretrain it from scratch. All models in our comparison (Mamba2, GDN etc.) were also trained from scratch in their respective papers.
> Moreover, our instation of CAT with dense attention admits a pure PyTorch implementation requiring no custom CUDA or Triton kernels, lowering the barrier to adoption relative to many efficient models.
> Further, CAT amortizes training cost by exposing multiple inference operating points from a single model -- training independent models for each would cost more overall (please see Sec. 4).

---

> > ### Author Rebuttal · Reviewer_F7bn · 2026-04-01
> >
> > Thanks for your response. Some questions are clarified, but considering the limited novelty and experimental deficiencies, I will keep my scores unchanged.

---

> > > ### Author Response · Authors · 2026-04-05
> > >
> > > We are glad our rebuttal clarified the reviewer's questions on algorithmic contributions and training costs.
> > >
> > > > re: limited Novelty
> > >
> > > Firstly, we don’t claim compression itself as a contribution. CAT's contribution is using compression in a simple way to enable test-time controllability of inference costs – a property absent in the works the reviewer mentioned and in existing sequence mixers. **This test-time controllability of inference costs in CAT is a novel contribution**, and CAT achieving better accuracy-inference costs trade-offs in a single model makes it a practical contribution too.
> > >
> > > > re: experimental deficiencies
> > >
> > > Our rebuttal clarified the validity of accuracy(capacity)-inference cost trade-offs as opposed to only parameter counts comparison, and discussed that CAT matches or surpasses models on long-context downstream tasks (real-world in-context recall task in Figure 1) when appropriately inference costs matched with other models.
> > >
> > > In contrast, table 6 and 7 provide a limited view on long-context abilities [1], focusing on averaged perplexity alone or small context downstream tasks (<30 tokens) [1]. We provided them for completeness, not as our main evaluations.
> > >
> > > We believe these address the experimental concerns raised in the initial review. Therefore, we are not sure what the reviewer means by “experimental deficiencies”. We would welcome concrete feedback on what “experimental deficiencies” refers to here, so we can address it directly.
> > >
> > > If the concerns are adequately addressed, we’d be grateful if the reviewer can re-consider the score for the paper.
> > >
> > > Thanks a lot in advance!
> > >
> > > [1] What is Wrong with Perplexity for Long-context Language Modeling?

---

### Official Review · Reviewer_FdrR · 2026-03-12

**Soundness:** 3
**Presentation:** 2
**Significance:** 2
**Originality:** 2
**Overall Recommendation:** 2
**Confidence:** 4

**Summary:**

This paper proposes a meta-sequence mixer: Compress & Attend Transformer (CAT). CAT decodes chunks of tokens by attending to compressed chunks of the sequence so far. Both compression and decoding from compressed chunks can make use of any existing sequence mixers.

**Compliance With Llm Reviewing Policy:**

Affirmed.

**Key Questions For Authors:**

see the weakness.

**Limitations:**

yes

**Strengths And Weaknesses:**

\+ Notably, compression results in decoding from a reduced sequence length that yields compute and memory savings,
while choosing a particular chunk size trades-off quality for efficiency. Importantly, training CAT with multiple chunk sizes at once, unlocks control of quality-efficiency trade-offs directly at testtime without any retraining, all in a single adaptive architecture.

\+ It demonstrates that CAT matches the dense transformer on language modeling while having lower inference costs (being 1.4 − 3× faster and using a 2 − 9× smaller total memory footprint). It can surpasses the dense transformer on real-world recall tasks using the most
performant setting (CAT-4) while still consuming lower inference costs (1.5× faster and 2× memory efficient), akin to MoEs.

\- The main paper mainly report the results of 1B model from the proposed method compared with smaller models such as Dense, Mamba2, GDN with 300M parameters, where it can lead to better accuracy performance. The results in Table 8 shows that compared with models of similar sizes, the accuracy is not significantly better, such as comparing with GDN-H 1:4 or GDN-H 1:1 on LongBench and SWDE. It seems that the method does not lead to better accuracy performance comparing with baselines of similar sizes.

\- The advantage of inference accleration and memory reduction does not seem to be signicant when comparing with models of similar size. for example, in table 8, GDN-H 1:4 can achieve 3.3x memory reduction and 2.4x speedup, whic is closer to CAT-4 with 4.2x and 2.9x, while the accuracy of  GDN-H 1:4 on some benchmarks are higher than CAT. It seems that the improvement of CAT is marginal compared with some baselines of the same size.

\-  CATS model takes about 2 times training FLOPs than much smaller dense baseline, with the help of FlexAttention API to obtain a custom self-attention kernel specifically for the masking scheme section 9. The paper experiments with small models. But if it is applied to larger models, it can be a big concern.

\- It  tests  language modeling and understanding on long contexts (upto 4K contexts). Since the motivation is long‑context efficiency, it is better to experiment with different configurations on the context length such as 8K, 16K.

---

> ### Author Rebuttal · Authors · 2026-03-31
>
> > W1
>
> Models spend more time doing inference than training in the age of agents.
> Therefore, for a model to be useful, what matters is the accuracy-inference cost trade-off the model provides. That is, how both (i) the model architecture and (ii) the parameters, interact with each other to give the final accuracy-cost trade-off.
> If a model can contain more parameters at similar inference cost due to its architecture design to ultimately provide a better trade-off – it is desired!
>
> MoEs exemplify this principle – it uses routing to allow more MLPs (parameters) to provide a better trade-off.
> So does CAT, with its particular design choice of compress then decode that allows using more parameters in the decoder to ultimately provide a better accuracy-cost trade-off.
>
> Further, to ensure fair and broad comparison, we changed configs (increased memory, parameters etc.) for different models in our comparison as long as their inference cost remained comparable to one of the CAT. This resulted in a total of 12 models, some of which have parameters comparable to CAT.
>
> CAT matches or surpass most models on long-context recall, owing to its **inference-first design** that allows for more parameters at lower costs, and thus resulting in a better accuracy-cost trade-off. Moreover, CAT provides a very desirable property absent in all models, changing of accuracy-cost trade-off at test-time without any re-training.
>
> For a discussion, we direct the reviewer to Sec. 3.1 and Sec. 4 of paper.
>
> > W1 contd: method does not lead to better accuracy comparing baselines of similar sizes.
>
> To help with an inference-matched comparison, we plot all models, including parameter matched CATs (in blue) on the accuracy-inference cost trade-off curve in the figure here: https://ibb.co/RGTJSxNY
>
> We observe GDN-H 1:4 and GDN-H 1:1 are not comparable to parameter matched CATs (blue) since they use more inference costs (GDN-H 1:1 uses upto 2x more memory and takes 1.6x longer for decoding).
>
> The right comparison to GDN-H 1:4 is CAT-8 (970M) (red dot), where CAT-8 (970M) provides a better trade-off.
>
> Further, while some models may have similar accuracy-cost trade-off as CAT, we argue the **primary goal of CATs is not to outperform them significantly, but to provide the flexibility to change the accuracy-inference cost trade-off at test-time.** No other approach currently provides this flexibility.
>
> Finally, we note that the strict competitors to CAT are sequence mixers with length-independent decoding costs (Mamba2 and GDN only) – since CAT reduces costs for any sequence mixer whose costs are length-dependent, specifically by reducing the sequence length using compression. Thus, **hybrids like GDN-H are actually complementary to CAT** -- refer to Sec. B.10 and B.11 where we use a GDN-H as a decoder in CAT). We nevertheless compared CAT with hybrids (across different hyper-parameters, configs, parameters) in our main tables for a broad evaluation.
> To drive this point home, we trained a CAT with GDN-1:1 as the decoder on 5B tokens of FineWeb on 1K context. We see CAT helps save cost in GDN, while even outperforming CAT that uses Dense, highlighting their complementary nature.
> | Model              | Mem. Savings ↑ | SWDE (recall) ↑ |
> |-|-|-|
> | Dense              | 1.0×          | 39.2          |
> | CAT-16 (Dense)     | 7×            | 13.5          |
> | GDN-H 1:1          | 2.5×          | 28            |
> | CAT-16 (GDN-H 1:1) | 14×           | 16.5          |
>
> We will make this clarification in the final draft.
>
> > W2
>
> Parameter matched CAT-4 (300M) (blue dot) uses ~22% lower memory than GDN-H 1:4 -- to put this in perspective, if GDN-H 1:4 would use 100GB GPU memory, then CAT-4 would use 78GB memory – this is enough of a difference to fit CAT-4 on a A100/H100 whereas GDN-H 1:4 would go OOM. As a result, we believe CAT-4 (300M) is not “close” to GDN-H 1:4 in inference costs.
>
> We highlight again that the right comparison to GDN-H 1:4 is CAT-8 (~970M), where CAT-8 provides a better trade-off.
>
>
> > W3
>
> We direct the reviewer to the discussion in Section 4 (specifically *What is the training cost of CATs?*)
>
> In short, at 4K context lengths, MLP costs dominate than attention costs – as a result, despite CATs using lower attention costs, they take 2x more FLOPs during pre-training. However, as pre-training moves to longer sequences e.g. 32K, this gap reduces to just 1.25x more FLOPs relative to the smaller dense transformer – this means CAT exposes four different models (with different inference cost) in a single model, just with 25% extra FLOPs during pre-training. These extra FLOPs continue to decrease as pre-training moves to even longer sequences in the near future (e.g. 1M tokens).
>
> **Note that pre-training multiple independent models from scratch that target exactly those inference costs would take significantly more FLOPs overall. As a result, CATs amortize the training cost of flexible inference.**
>
> > W4
>
> Please refer to the response to reviewer `5rjE`

---

> > ### Author Rebuttal · Reviewer_FdrR · 2026-04-02
> >
> > Thanks for the rebuttal. Some of my concerns are addressed. But I still have some concerns on experiment setting. It is better to update the experiments in the main paper. I think it needs significant update for the current paper.

---

> > > ### Author Response · Authors · 2026-04-05
> > >
> > > We are happy that some concerns have been addressed.
> > >
> > > > It is better to update the experiments in the main paper. I think it needs significant update for the current paper.
> > >
> > > The figure we provide above in our rebuttal is directly taken from Table 8, which will be added in the appendix (we did not run any new experiment for this).
> > >
> > > The new evidence we provide for the complementary nature of hybrids and CAT, and for long-context (8K) evaluations are two tables which can be incorporated in the additional page that ICML provides.
> > >
> > > The discussion on inference-cost-matched comparison appears in Sections 3.1 and 4 -- we will sharpen the wording in the final draft.
> > >
> > > As a result, we respectfully note that these do not constitute significant changes to the paper.
> > >
> > > > But I still have some concerns on experiment setting
> > >
> > > We believe we have addressed each experimental concern raised in the initial review, namely:
> > > - clarified the validity of inference costs matched comparison among models. Parameter matched CATs (blue dots in this figure: https://ibb.co/RGTJSxNY) consume lower inference costs than all models in comparison, and hence are not the right comparison -- we provided those results for completeness and transparency, not as comparison
> > > - for a fair evaluation, we exhaustively changed configs for models (increased state-size, parameters, change hybrid ratios etc.) as long as their costs remained comparable to a CAT
> > > - importantly clarified that **primary goal of CATs is not to outperform every model significantly**, but to provide the *flexibility to change the accuracy-inference cost trade-off* at test-time. **No other approach currently provides this flexibility.**
> > >
> > >
> > > and provided additional evidence in our rebuttal for CAT -- specifically:
> > > - complementary nature with hybrids
> > > - long-context 8K (and now 16K in [response](https://openreview.net/forum?id=QArodlQFBQ&noteId=j7vgYV3wqJ) to reviewer `5rjE `) evaluations
> > >
> > > We would be very happy to address any remaining concerns -- if the reviewer could specify exactly which aspects of the experimental setting they still find insufficient, we will address them in the final version.
> > >
> > > If the concerns are adequately addressed, we’d be grateful if the reviewer can re-consider the score for the paper.
> > >
> > > Thanks a lot in advance!

---

### Official Review · Reviewer_5rjE · 2026-03-12

**Soundness:** 2
**Presentation:** 3
**Significance:** 1
**Originality:** 2
**Overall Recommendation:** 3
**Confidence:** 4

**Summary:**

The authors propose a chunked compress-then-decode architecture for efficient language modeling. The approach is to compress token chunks into shorter representations, then decode autoregressively over those compressed chunks. Here, chunk size becomes a tunable parameter that can act as an immediate quality vs. efficiency tradeoff at test-time.

**Compliance With Llm Reviewing Policy:**

Affirmed.

**Final Justification:**

I appreciate the authors’ responses and the additional results. The paper is built on a strong idea, but it still needs revision in its current form to better establish novelty and isolate gains from parameter scale.

**Key Questions For Authors:**

1. What happens at larger context lengths, e.g. 64K, 128K and beyond?
2. How does the approach compare directly to BLT and similar byte / latent compression models?
3. Could the authors test adaptive compression and experimentally validate if it improves large-chunk performance?
4. How does the method perform on more current reasoning, instruction-following, or code benchmarks such as GSM8K, MATH, GPQA, IFEval, Arena-Hard, HumanEval, LiveCodeBench, or SWE-bench Verified, especially in long-context scenarios?

**Limitations:**

yes

**Strengths And Weaknesses:**

Strengths:
- Useful problem. Test-time control from a single trained model is very useful.
- Paper is well written and clear.
- The approach design, graphics, implementation, and evaluations are easy to follow.
- Baseline comparisons support a strong Pareto evaluation.

Weaknesses:
- Baseline evaluation stops at 4K context and do not effectively test the long-context claim.
- Prior work discussion is incomplete. Need direct comparison to byte / latent compress then decode models (e.g. Byte Latent Transformer (BLT), Pagnoni et al., ACL 2025; DOI: https://doi.org/10.18653/v1/2025.acl-long.453).
- Fixed one vector per chunk compression becomes a bottleneck at larger chunk sizes. This is only briefly acknowledged during discussion.
- Real-world evaluation remains limited. No strong reasoning, instruction-following, or code-generation evaluation.

---

> ### Author Rebuttal · Authors · 2026-03-30
>
> > W1 and Q1
>
> We provide scaled up results on 8K sequence length.
> All models were trained from scratch on 8K sequence length for 30B tokens on FineWeb-Edu.
> We additionally increased the number of layers to 18 for all models.
>
> | Model   | Memory Savings ↑ | NIAH-N 2K | NIAH-N 4K | NIAH-N 8K | Avg. LM evals |
> |---------|---------------|-----------|-----------|-----------|---------------|
> | Dense   | 1×            | 99.3      | 59.5 | 37.9 | 44            |
> | GDN-1:1 | 4×            | **99.7**  | 53.2      | 24.9      | 44.4          |
> | CAT-4   | 2×            | 99 | **90.4**  | **98**    | 44.6          |
> | CAT-8   | 3.8×            | 94.4      | 92        | 87        | 44.5          |
> | CAT-16  | 6.2×            | 82.7      | 82.7      | 57.8      | 45.3   |
> | CAT-32  | 10×           | 71.8      | 42.9      | 26.2      | **45.8**      |
>
> CAT performs strongly especially at a longer context of 8K, providing a better accuracy-cost trade-off.
> Further, note that all CATs reported above are a single model, where inference costs can be changed depending on the downstream task.
>
> > W2 and Q2
>
> BLT [3] utilizes an “hourglass” like model architecture, where chunks are compressed/downsampled using a local encoder. The chunk boundaries are according to entropy judged by another auxiliary model (~100M parameters) that requires separate training. These compressed chunk representations are then fed to a latent transformer that operates on these chunk representations. These compressed chunk representations are then upsampled again to the original sequence length and then decoded back by a local decoder. Due to the hourglass structure (same as [1, 2]), there are compute savings during training, but during generation, the architecture must maintain a cache for all the past tokens, leading to significant memory accesses (primary bottleneck in decoding) in the local decoder.
>
> The goal of BLT is to try to compress in an “data-dependent” way according to entropy.
> **The goal of CAT is actually complementary: it is to provide controllable efficiency at test-time.**
>
> **BLT does not provide controllable efficiency. If the same sequence is fed to BLT, there is no mechanism to change compute/memory for that sequence in BLT.**
>
> Further, one can instantiate the main network of BLT as a CAT sequence mixer, and get both data dependent compression and controllable efficiency.
>
> We will cite the work and update the discussion in related work.
>
> [1] SpaceByte: Towards Deleting Tokenization from Large Language Modeling
>
> [2] Efficient transformers with dynamic token pooling
>
> [3] BLT, ACL 2025
>
> > W3 and Q3
>
> Before we arrived at the current design, we explored various approaches including compressing into a set of vectors (“registers”) and then decoding from those vectors.
>
> However, this approach did not work well empirically – in-context recall performance was inferior and it was sensitive to hyper-parameters (e.g. LR) on the MQAR task (same task used in Fig. 4).
>
> As described in Section 4 (*Empirically, we observed that wider decoder…*), increasing dimensionality was the only thing that worked well (see Sec. D.2) – perhaps due to a fundamental information theoretic result relating available information in compressed representations and compute required to decode from that representation [1].
>
> With increased dimensionality of the decoder, decoding from a single representation worked better than decoding from two vectors but using a lower dimensionality decoder.
>
> One could try having multiple vectors with an increased decoder dimensionality and that may give better quality at the cost of increased inference costs (KV cache would increase 2x due to 2x compressed vectors now). However, we note that decoding from a single compressed representation simplifies the design. We leave such exploration to future work.
>
> *That being said, if one wishes to lower compression and increase quality, one can just change the knob (chunk size) to use CAT-16 at test-time.* This is a feature unique to CAT.
>
> [1] A theory of usable information under computational constraints. arXiv:2002.10689
>
>
> > W4 and Q4
>
> We test all models on standard pre-training benchmarks used by numerous previous papers [1, 2, 3, 4]: including common-sense reasoning (Table 7), real-world in-context recall (Table 1), long-context modeling (Table 2, 3).
>
> The benchmarks mentioned by the reviewer (GSM8K, MATH, GPQA, SWE-bench etc.) require training substantially larger model and data scales to yield non-trivial performance – which is not possible for us with limited compute. [1, 2, 3, 4] does not evaluate on these benchmarks either.
>
> We hope the benefits demonstrated here (KV cache reduction, test-time control of efficiency, good in-context recall), on this scale should transfer directly to larger models, as evidenced when [1, 3] were scaled up.
>
> [1] Gated Delta Networks ICLR 2025
> [2] Forgetting Transformer ICLR 2025
> [3] Mamba2, ICML 2024
> [4] GLA, ICML 2024

---

> > ### Author Rebuttal · Reviewer_5rjE · 2026-04-01
> >
> > I thank the authors for the response and their efforts. The 8K result partially addresses W1, and the clarification on BLT is reasonable for W2. However, my main concerns remain. The new evidence is still limited to 8K, while my question was about 64K+ settings, where practical memory savings would require larger chunk sizes and the issue in W3 is more likely to appear. CAT-16 already drops to 57.8% at 8K, which makes this limitation more relevant. The rebuttal also does not explore alternatives such as within-sequence adaptive chunking. One  question is whether the CAT-4 gain from 4K to 8K (90.4 to 98.0) is stable across seeds and whether it also holds at 16K. At present I will maintain my current score.

---

> > > ### Author Response · Authors · 2026-04-05
> > >
> > > > practical memory savings would require larger chunk sizes
> > >
> > > We'd like to clarify a potential confusion: The reviewer feels that at large sequence lengths, one requires large chunk sizes to have memory savings -- this is not true.
> > >
> > > CAT provides memory savings at **any** chunk size relative to a dense transformer at **any** sequence length.
> > >
> > > Compression in CAT (controlled by chunk size) is independent of sequence length. The number of compressed chunks simply grows as the sequence length increases, so there is no hard fixed memory in CAT. Please refer to Table 4 for comparison to related work where CAT provides both **flexible** and **efficient memory** usage, providing a **middle ground** compared to the *full memory* dense transformer, and *hard fixed* memory bottlenecks like linear attention.
> > >
> > > The memory in CAT grows as sequence length increases, it just grows *gracefully* by a constant factor less than the dense transformer, providing memory savings at any sequence length (refer to Figure 2 in the main paper).
> > >
> > > This means, if one chooses to operate with CAT-4 or CAT-8 or CAT-16 at longer sequences, one would have 2x or 4x or 8x memory reduction compared to a dense transformer always.
> > >
> > > If the reviewer feels 2-8x memory reduction is not practical at any sequence length (including 64K), we’d be happy to hear their perspective.
> > >
> > > Further, CAT allows one to seamlessly switch between these memory reductions (2x or 4x or 8x) at test-time.
> > >
> > > > The new evidence is still limited to 8K, while my question was about 64K+ settings
> > >
> > > Training a model at 64K is beyond our compute budget.
> > >
> > > However, we do provide new evidence at 16K sequence length. We perform continued pre-training for all models for 1B tokens at 16K sequence length.
> > >
> > > | Model   | 2K     | 4K     | 8K     | 16K    |
> > > |---------|--------|--------|--------|--------|
> > > | Dense   | 99.3   | 59.5   | 37.9   | 5      |
> > > | GDN-H   | 99.7   | 53.2   | 24.9   | 13.2   |
> > > | CAT-4   | 99     | 90.4   | 98     | 90.1   |
> > > | CAT-8   | 94.4   | 92     | 87     | 82.05  |
> > > | CAT-16  | 82.7   | 82.7   | 57.8   | 51.8   |
> > >
> > > > the issue in W3 is more likely to appear. CAT-16 already drops to 57.8% at 8K, which makes this limitation more relevant
> > >
> > > We observe that while all models drop at longer sequences, the drop in CAT-16 is much more graceful at 16K, providing better quality at the same or lower inference costs.
> > >
> > > CAT-4/8 provide >80% accuracy at 16K and provide 2-4x memory savings respectively compared to the dense transformer.
> > >
> > > Crucially, CAT allows switching between these different accuracy-cost points at test time depending on the downstream task. **This controllable efficiency from a single model is CAT's primary goal, which we provide exhaustive evidence for.** Improving long-context performance beyond what is possible right now in CAT is interesting future work.
> > >
> > > > within-sequence adaptive chunking
> > >
> > > CAT uses the same chunk size for the entire sequence, which is changeable at test time. This change of chunk size at test-time provides controllable efficiency which is CAT’s primary goal.
> > >
> > > Varying chunk sizes within a single sequence (which is what BLT provides) is an interesting complementary direction that we discussed above, but falls outside the scope of this work, and we leave this to future work.
> > >
> > > > stable across seeds
> > >
> > > We provide results from a different seed for CAT-4
> > >
> > > | Model | 2K   | 4K   | 8K   |
> > > |-------|------|------|------|
> > > | CAT-4 | 98.3 | 95.0 | 96.7 |

---

### Official Review · Reviewer_Wpus · 2026-03-13

**Soundness:** 2
**Presentation:** 2
**Significance:** 2
**Originality:** 2
**Overall Recommendation:** 4
**Confidence:** 2

**Summary:**

The paper presents a way to do efficient decoding of LLMs. Instead of attending to every token in the past KV cache, the paper suggests first using a trained compressor to periodically compress the past block of tokens into a single compressed token. Then, the paper proposes to train the decoder to take in the past compressed tokens and generate autoregressively the next token. The paper compares the given method with past efficient attention methods and show that on long-context recall tasks alone, the method proposed CAT is able to surpass previous Sparse Transformer and past linear attention methods.

**Compliance With Llm Reviewing Policy:**

Affirmed.

**Final Justification:**

I think the paper presents a valuable study into the design of attention KV compression. The project requires a huge amount of resources to meaningfully scale up to a 1B model parameter size. The authors showcase strong recall performance on the performance of the architecture. However, the paper initially contains minor false claims (pointed out in the review) and also put important general benchmark scores toAppendix, which hurts the paper's presentation. Given that the paper hugely modifies the conventional model architecture, I suggest that the authors study the mid-training and posttraining dynamics and study the downstream tasks' performance going forward.

**Key Questions For Authors:**

1. How does proposed architecture perform on general language modeling tasks. The multiple-choice questions, etc.
2. How does the proposed method perform against the block-sparse attention baseline which are training-free and also significantly reduce the decoding computation.

**Limitations:**

Yes

**Strengths And Weaknesses:**

Strengths:
1. The paper proposes a novel method that does efficient decoding, while study the various design compress rates. The study is beneficial to the efficient attention community.
2. The paper take solid considerations over the past competing methods. The paper faithfully pretrain the prior methods to illustrate the models' strong long-context performance.

Weaknesses:
1. I found the following important claim in the paper really confusing. In the paper, it mentions that when the decoder is outputting the next token, it looks at the current window and all the past compressed tokens. Then considering the prefill token to be N, and by definition, there are (N/C) compressed tokens, then the total cost of decoding the next C tokens is O((N/C + N/C + C) * C * 1/2), which is O(N + C^2), while the normal attention is O(NC + C^2). Since the paper targets long-context decoding, N >> C. I found the statement in the paper that claims the decoding complexity drops from O(N^2) to O(N) to be very confusing.
2. Since the paper requires pretraining, the architecture is highly different from normal attention. Evaluating long-context performance only isn't enough to showcase the practical value of the special design. Normal pretraining metrics are important and missing from the paper's evaluations.
3. An important baseline I am interested in seeing, which isn't considered yet by the paper, is the vanilla training-free block-sparse attention. A reference implementation can be found in the following paper (simple, chunk the past KV into blocks, and then do Top-K to select the past blocks). It is different from the paper's design, but I think the block-sparse is training-free, requires light implementation considerations, and can be directly applied to the baseline model used by the paper. I think analysis of the cost and performance (also scalability of the compression ratio) can be a valuable additional analysis of the proposed method.
Sun, H., Chen, Z., Yang, X., Tian, Y., & Chen, B. (2024). Triforce: Lossless acceleration of long sequence generation with hierarchical speculative decoding. arXiv preprint arXiv:2404.11912.

---

> ### Author Rebuttal · Authors · 2026-03-30
>
> > W1
>
> The reviewer is right in their analysis for decoding. However, we’d like to point out that we don’t mention in the paper that decoding complexity drops to O(N) for CAT.
>
> We only said that compression of chunks is O(N*C) since it happens in parallel across chunks with a maximum context length of C (this is the prefill step) across all chunks, at lines 178-180.
>
> The decoding after the prefill step is still quadratic in complexity, just that it can be a constant times better than the dense transformer depending on the chosen chunk size.
>
> We further note that decoding is extremely memory access bound – it takes much more time to transfer KV cache from global memory (HBM) to on-chip memory (SRAM), than to perform actual attention computation. As a result, CAT reduces this transfer time due to significantly reduced KV cache, which results in large speed ups in decoding depending on the chunk size (Figure 2).
>
> We will clarify this in the final draft.
>
> > W2 and Q1
>
> We report standard pre-training benchmarks in Table 7. We note that CAT performs competitively on all datasets, outperforming most models while allowing for a controllable inference cost at test-time -- something that no other model provides.
>
> > W3 and Q2
>
> We thank the reviewer for bringing this to our notice.
>
> We’d like to point out that Triforce results in reduced latency, however, it does not result in memory savings since one still needs to keep the entire KV cache around (for the target model).
>
> Note that memory is the major bottleneck during decoding, especially at long-contexts (GPU memory runs out faster than compute).
>
> Further, we believe that all post-hoc methods (like Triforce, or Speculative decoding) are complementary and can be applied to any sequence mixer out there: Attention, Mamba2, GDN, including CAT. As a result, we believe these are not baselines (we further direct the reviewer to Table 4 in the appendix for a comparison with closely related works).
>
> CAT at its core is a meta-sequence mixer, where compressor/decoder can use any sequence mixer of choice. In our experiments, we deliberately used the simple dense transformer as the decoder since dense attention is so ubiquitous – hence, Triforce/Speculative Decoding can be easily applied in the decoder resulting in additional compute savings, besides what CAT already provides.
>
> We will cite and add a discussion about this in the paper.

---

> > ### Author Rebuttal · Reviewer_Wpus · 2026-04-04
> >
> > Hi, thank you for rebuttal. I appreciate the authors' investment in studying the topic. However, I still found that block-sparse attention should be an important baseline to compare against. Also, I think it cannot be readily applied to the linear attention line of work since they are usually constant-KV requiring no selection of past tokens. Therefore, I maintain my score.

---

> > > ### Author Response · Authors · 2026-04-05
> > >
> > > > I still found that block-sparse attention should be an important baseline to compare against.
> > >
> > > We’d like to point out that training-free block-sparse attention **does not result in memory savings**, it **only results in compute savings**.
> > >
> > > Please refer to table 4 where we discuss the closely related works and discuss how CAT satisfies both compute and memory savings during inference.
> > >
> > > As a result, it's not a baseline that CAT should be compared against. In fact, it is complementary to CAT as we discussed in our rebuttal earlier (assuming CAT's decoder uses a sequence mixer that has length dependent costs -- that's why we use vanilla attention in CAT. For more discussion, refer to our response later in the reply).
> > >
> > > Nevertheless, since the reviewer feels its important, we evaluate training-free block-sparse attention (as the reviewer suggested: chunk the past KV into blocks, and then do top-K to select the past blocks) taken from the MInference 1.0 method [1] using the reference implementation from `nano-sparse-attention` repo [2]. We evaluate on the SWDE recall task.
> > > We use `block_size = 32`, and vary the top-K parameter.
> > >
> > > It won’t be possible for us to conduct a latency benchmark for training-free block-sparse attention in this limited rebuttal time since it requires setting up proper infrastructure setup (e.g. setting up efficient top-K selection under CUDA graphs etc.), we provide theoretical per-token FLOPs reduction and theoretical memory reduction for both CAT and sparse-block attention. We also include trainable strided sparse attention models [3] in this comparison that we already trained in the paper in Table 1.
> > >
> > > We calculate per-token FLOPs savings for block sparse attention as `O(N / (block_size * top_K))` where sequence length `N = 4096` tokens, and `block_size = 32` tokens (as discussed above). For CAT, we use `O(N / (chunk_size))` as FLOPs per-token, and use `O(N / (chunk_size))` as theoretical memory reduction.
> > >
> > > ||theoretical memory red.|theoretical per-token FLOPs red.|SWDE|
> > > |-|-|-|-|
> > > |Dense|1x|1x|43.4|
> > > ||||
> > > |**Block Sparse Attention (Training free)**||||
> > > |top-32|1x|4x|43|
> > > |top-16|1x|8x|34.4|
> > > |top-8|1x|16x|18.9|
> > > |top-4|1x|32x|8.0|
> > > ||||
> > > |**Trained Strided Sparse Attention**||||
> > > |Sparse-4|4x|4x|36|
> > > |Sparse-8|8x|8x|20.9|
> > > ||||
> > > |**CATs**||||
> > > |CAT-4|4x|4x|47.1|
> > > |CAT-8|8x|8x|36.5|
> > > |CAT-16|16x|16x|21.5|
> > > |CAT-32|32x|32x|8.2|
> > >
> > >
> > > Training-free block sparse attention **only** provides FLOPs savings, **it does not provide memory savings**.
> > >
> > > **CAT provides BOTH compute and memory savings, and at a better recall-cost trade-off** – owing to its architecture design and end-to-end training of compression and decoding.
> > >
> > > *Moreover, we emphasize that in long-context scenarios today, memory savings are much more important than FLOPs savings, since the current hardware provides more FLOPs than GPU memory.*
> > >
> > >
> > > > I think it cannot be readily applied to the linear attention line of work since they are usually constant-KV requiring no selection of past tokens
> > >
> > > The reviewer is correct that the post-hoc method block-sparse attention can only be applied to sequence mixers whose decoding costs are sequence length dependent – that is, it can be applied to attention, but it cannot be applied to linear attention (pure Mamba2, pure GDN) since decoding cost of linear attention is independent of the sequence length.
> > >
> > > CAT as a meta-sequence mixer saves costs only for sequence mixers whose costs are length dependent (i.e. attention, or hybrids -- which are models being currently deployed in production).
> > >
> > > CAT cannot save costs for linear attention (e.g. pure GDN, pure Mamba2) and hence they are the only strict baselines to CAT (this is discussed in our related work Sec. 5, lines 412 in the paper).
> > >
> > > Rest of the sequence mixers (attention, hybrids) or post-hoc techniques (block-sparse attention, spec. decoding) are complementary to CAT, and can be mixed-and-matched. We show this complementary nature of CAT to hybrids in response to reviewer `FdrR`, and in appendix Sec. B.11.
> > >
> > > **Further, we deliberately used attention as both compressor/decoder in CAT due to its ubiquitous nature, and any post-hoc technique developed for attention (block-sparse attention, speculative decoding etc.) presently or in the future can be readily applied to CAT.**
> > >
> > > We further show that CAT with the most basic choice of attention suffices to achieve a pareto-frontier in recall-inference costs trade-offs compared to existing complicated sequence mixers.
> > >
> > >
> > > Thanks a lot for your time! and we hope this addresses any remaining concerns.
> > >
> > >
> > > [1] MInference 1.0: Accelerating Pre-filling for Long-Context LLMs via Dynamic Sparse Attention
> > >
> > > [2] https://github.com/PiotrNawrot/nano-sparse-attention
> > >
> > > [3] Generating Long Sequences with Sparse Transformers, OpenAI

---

### Decision · Program_Chairs · 2026-04-30

**Decision:**

Reject

**Comment:**

The work studies a meta-sequence mixer called Compress & Attend Transformer (CAT) that utilizes chunk-based compression to provide test-time controllability of inference costs.

The scores are trending negative (4, 3, 3, 2).

Reviewer Wpus, the only reviewer with a positive score indicated "I still found that block-sparse attention should be an important baseline to compare against. Also, I think it cannot be readily applied to the linear attention line of work since they are usually constant-KV requiring no selection of past tokens. "

A few concerns remained after the rebuttal including capacity mismatch, novelty, long context bottleneck, and experiments.

Therefore the AC finds that the work is not ready to be published at its current state.